# Learning Gaussian DAG Models without Condition Number Bounds

**Constantinos Daskalakis** [* 1]  **Vardis Kandiros** [* 1]  **Rui Yao** [* 1]

## Abstract

We study the problem of learning the topology of a directed Gaussian Graphical Model under the equal-variance assumption, where the graph has $n$ nodes and maximum in-degree $d$. Prior work has established that $O(d \log n)$ samples are sufficient for this task. However, an important factor that is often overlooked in these analyses is the dependence on the condition number of the covariance matrix of the model. Indeed, all algorithms from prior work require a number of samples that grows polynomially with this condition number. In many cases this is unsatisfactory, since the condition number could grow polynomially with $n$, rendering these prior approaches impractical in high-dimensional settings. In this work, we provide an algorithm that recovers the underlying graph and prove that the number of samples required is independent of the condition number. Furthermore, we establish lower bounds that nearly match the upper bound up to a $d$-factor, thus providing an almost tight characterization of the true sample complexity of the problem. Moreover, under a further assumption that all the variances of the variables are bounded, we design a polynomial-time algorithm that recovers the underlying graph, at the cost of an additional polynomial dependence of the sample complexity on $d$. We complement our theoretical findings with simulations on synthetic datasets that confirm our predictions.

## 1. Introduction and Background

A common problem that arises in many scientific disciplines is inferring the underlying dependency structure of a vector of observations (Duncan, 2014). In particular, this is one of the fundamental goals of causal inference, where the emphasis is on discovering cause-and-effect relationships between variables. Perhaps the most popular way of approaching this problem is through the formalization of **Directed Acyclic Graph (DAG)** models, or **Bayesian Networks**, where variables are sampled sequentially, with the order induced by a DAG $G$ (Lauritzen, 1996; Wainwright et al., 2008). These models have been widely adopted in the social sciences, starting from their introduction in the seminal work (Pearl et al., 2000). They have also found applicability in fields as diverse as human geography (Malioutov et al., 2006; Ao & Chang, 2020), the environment (Grace et al., 2007), and ecology (Byrnes et al., 2011). The problem of determining $G$ from observations of the variables is referred to in the literature as **causal structure learning**.

In the general case, some popular approaches for solving this problem include the PC algorithm (Spirtes et al., 2001) and Greedy Equivalence Search (GES) (Chickering, 2002b). However, these often require strict assumptions, such as strong faithfulness (Uhler et al., 2013), or encounter computational issues. Indeed, in general, the problem of recovering the true Bayesian Network is NP-hard (Chickering, 1996).

The problem becomes more structured using the formalization of Structural Equation Models (SEMs), where each variable is equal to a function of its parents with some added independent noise. Identifiability for these models has been established when the functions are sufficiently non-linear or when the noise is non-Gaussian (Peters et al., 2014; Shimizu et al., 2006). For linear Gaussian SEMs, which is the focus of this work, finding the graph in general is impossible. Identifiability was established in (Peters & Bühlmann, 2014), under the additional assumption that all $\varepsilon_i$ have the same variance (**equal variances assumption**). A subsequent flurry of works (Ghoshal & Honorio, 2017b;a; 2018; Chen et al., 2019; Gao et al., 2020; 2022) established upper and lower bounds on the number of samples needed (**sample complexity**) for inferring the graph in the high dimensional sparse setting where the maximum in-degree of any node is bounded by $d$ and the number of nodes $n >> d$. Often, the goal in these works is to obtain sample complexity that scales as $O(d \log n)$.

However, an important factor that is often overlooked in

---

[*]Equal contribution [1]Department of EECS, Massachusetts Institute of Technology, Cambridge, Massachusetts, United States. Correspondence to: Constantinos Daskalakis <costis@csail.mit.edu>, Vardis Kandiros <kandiros@mit.edu>, Rui Yao <rayyao@mit.edu>.

*Proceedings of the 42$^{nd}$ International Conference on Machine Learning*, Vancouver, Canada. PMLR 267, 2025. Copyright 2025 by the author(s).

prior work is the role of the condition number $\kappa$ of the covariance matrix of observations. Indeed, all previous analyses typically assume some type of condition that implies a bound on $\kappa$, independent of $n$. However, in many interesting examples and simple examples, $\kappa$ scales polynomially with the number of nodes $n$. This would imply that all algorithms in prior work require $\text{poly}(n)$ samples, which is problematic in high-dimensional settings. For example, in the directed path graph or the tree graph, $\kappa$ grows polynomially with $n$, but the structure can clearly be identified easily using $O(\log n)$ samples. Thus, a natural question that arises is whether it is even ***information theoretically*** possible to design an algorithm that is truly high-dimensional, with a sample complexity independent of the condition number.

In this work, we answer this question in the affirmative. In particular, we design and analyze an algorithm that achieves graph recovery with sample complexity independent of the condition number while at the same time retaining the optimal $O(d \log n)$ scaling. We identify a key quantity $\tau(G)$ of the graph (see (2)) that determines this sample complexity, which is always upper bounded by the condition number $\kappa$. Furthermore, we provide a lower bound for the sample complexity of any algorithm, which also contains $\tau$ and matches our upper bound up to a single factor of $d$ (this $d$-factor discrepancy of the upper and lower bound is a known limitation of all prior lower bound approaches for Markov Random Fields (Wang et al., 2010; Misra et al., 2020; Kelner et al., 2020)). Thus, we obtain an almost tight characterization of the optimal sample complexity of the problem. Another natural question is whether we can design an efficient algorithm that solves the problem without condition number assumptions. To this end, we provide such an algorithm, which works under the natural assumption that the maximum variance of a node is bounded by a constant. We also show that this is a strictly weaker assumption than the boundedness of the condition number.

In terms of techniques, the problem of learning a DAG is information-theoretically different in nature from that of learning undirected graphical models due to the combinatorial nature involved in finding the right topological ordering of the nodes. Despite that, our approach brings together a variety of tools from the literature on learning directed as well as undirected graphical models. We hope this serves as an inspiration for further fruitful investigations that would uncover connections between the two problems.

The rest of the paper is organized as follows. Section 1.1 discusses prior work related to the problem, and Section 1.2 contains some preliminary definitions and facts about the model. The main Algorithm that achieves information-theoretic discovery is given in Section 2.1, with the matching lower bound following in Section 2.2. An efficient Algorithm is presented in 2.3. For completeness, we provide a more detailed technical comparison of our results with prior work in Section 2.4. Finally, we give a proof sketch for the main results in Section 3 and simulation results in Section 4.

## 1.1. Prior work

There is a long line of work that focuses on recovering the underlying directed acyclic graph (DAG) in structural equation models. A popular approach is the PC algorithm (Spirtes et al., 2001), which runs in polynomial time if the graph is sparse. (Robins et al., 2003) shows that it is point-wise but not uniformly consistent under the classical faithfulness assumption on the underlying distribution. (Zhang & Spirtes, 2012; Kalisch & Bühlman, 2007) show uniform consistency under the strong faithfulness assumption, which can be sometimes restrictive (Uhler et al., 2013). Other approaches include score-based model selection(Spiegelhalter et al., 1993; Heckerman et al., 1995; Chickering, 2002b;a), where searching for a graph can be cast as an optimization problem.

Closer to our work is a line of research that considers learning ***structural equation models*** (SEMs). Various works show identifiability when the functions are sufficiently non-linear (Peters et al., 2014) or when they are linear and the noise is non-Gaussian (Shimizu et al., 2006). In general, when the noise is Gaussian, it is impossible to determine the direction of the edges, which makes the problem non-identifiable. (Peters & Bühlmann, 2014) proved identifiability under the equal variances assumption, followed by an investigation on the optimal sample complexity (Ghoshal & Honorio, 2017a;b; Chen et al., 2019; Gao et al., 2020; Gao & Aragam, 2021; Gao et al., 2022). In particular, in (Gao et al., 2022) they provide an algorithm that requires $O(d \log n)$ samples. As we mentioned earlier, all these prior analyses assume that the condition number is bounded, and the sample complexity is implicitly dependent on that with a polynomial rate. Another notable work is (Gao et al., 2023). This paper focuses on support recovery in a linear regression setting where the underlying design satisfies additional structural constraints. The authors provide an alternative approach to Best Subset Selection and show it performs better in a setting where there is no generalized path cancellation.

Finally, another popular branch of the literature considers learning undirected Markov Random Fields (MRFs) (Bresler, 2015; Vuffray et al., 2020; 2016; Lokhov et al., 2018; Klivans & Meka, 2017; Wu et al., 2019; Misra et al., 2020). In general, this problem possesses an inherent convexity and can be solved by neighborhood regression, which is why it is considered easier than the directed case. In the special case where the underlying distribution is Gaussian, the so-called undirected ***Gaussian Graphical Model*** (GGM), a variety of algorithms have been proposed when

the underlying graph is sparse, based on estimating the co-variance matrix(Meinshausen & Bühlmann, 2006; Yuan & Lin, 2007; Cai et al., 2011; Anandkumar et al., 2012; Cai et al., 2016; Johnson et al., 2012; Friedman et al., 2008). Under minimal assumptions, (Wang et al., 2010) proved a sample complexity lower bound of $\Omega((1/\beta_{\min})\log n)$, where $\beta_{\min}$ is the minimum normalized edge strength of the model. (Misra et al., 2020) provides an algorithm that succeeds with $O((d/\beta_{\min})\log n)$ samples, and (Kelner et al., 2020) designs a computationally efficient alternative with similar sample complexity under the additional assumption of walk-summability (Malioutov et al., 2006). An interesting folklore question is to compare the complexity of the problems of identifying the undirected and directed graphs of Gaussian models(Gao et al., 2022). Our work can thus be seen as establishing a separation between these two tasks, which are governed by different quantities, $\beta_{min}$ in the case of undirected and $\tau$ in the case of directed models. Finally, a related line of work studies Gaussian models with the additional assumption that the topology is a tree, see e.g.(Tan et al., 2010; 2011).

## 1.2. Preliminaries and Problem Settings

Let $G = (V, E)$ be a Directed Acyclic Graph (DAG) on $n$ nodes, where the maximum in-degree of a node is bounded by $d$. For $i, j \in V$, we sometimes write $i \to j$ if $(i, j) \in E$ is a directed edge. As discussed in Section 1, $G$ can be thought of as the fixed ground truth graph that we seek to identify. We denote by $\text{pa}(i) = \{j \in V : j \to i\}$ the parent-set of node $i$. By assumption, $|\text{pa}(i)| \leq d$ for all $i \in V$. A ***topological order*** of the nodes in $G$ is any permutation of the nodes $\pi : V \mapsto [n]$ such that if $i \to j$ implies $\pi(i) < \pi(j)$. Since $G$ does not have directed cycles, this ordering always exists but is not necessarily unique. With each node $i$ in $G$, we associate a random variable $X_i \in \mathbb{R}$. A vector of random variables $(X_1, X_2, \ldots, X_n) \in \mathbb{R}^n$, is generated according to DAG $G$ if the following structural equation holds

$$X_i = \sum_{j \in \pi(i)} b_{ji} X_j + \varepsilon_i, \tag{1}$$

where $\varepsilon_i \sim \mathcal{N}(0, \sigma^2)$ is i.i.d. Gaussian noise of equal variance. In particular, if a node $i$ has no parents, $X_i$ is just $\mathcal{N}(0, 1)$. The above model is the so-called ***equal variances model***. We note that when this assumption is not satisfied, it might be impossible to identify the DAG, even up to the Markov Equivalence Class. For details see E.1. For a set $I$, we denote $X_I$ to be the vector of $(X_i)$ for $i \in I$.

We can collect all coefficients $b_{ji}$ in a matrix $B \in \mathbb{R}^{n \times n}$, so that $B_{ij} = b_{ij}$ is non-zero only if $i \to j$. If we index the rows and columns of $B$ according to a topological ordering, we can think of $B$ as being upper triangular, and the

following relation holds

$$X = B^\top X + \sigma\epsilon,$$

where $X$ is the vector of random variables indexed by the same topological order, and $\varepsilon$ is a standard $n$-dimensional Gaussian vector. This has the straightforward implication that the covariance $\Sigma$ and inverse covariance $\Theta$ of vector $X$ can be written as

$$\Theta = \sigma^{-2}(I-B)(I-B)^\top, \Sigma = \sigma^2((I-B)^{-1})^\top(I-B)^{-1}$$

It is clear that $\sigma$ simply controls the scale of all variables in the problem and doesn't affect structure recovery. Therefore, for simplicity in the remainder of this work, we assume that $\sigma = 1$. If $\sigma$ was unknown, our results extend straightforwardly. Algorithm 1 has the same sample complexity but with an added first step to estimate the smallest variance. The number of samples in Algorithm 2 would scale by $\sigma^{-2}$. More details can be found in Supplementary E.3. We make the following assumption throughout this work.

**Assumption 1.1.**

$$\min_{i \in V, j \in \text{pa}(i)} |b_{ji}| \geq b_{\min} > 0$$

Assumption 1.1 places a lower bound on the minimum strength of an edge between two nodes. This lower bound is essential information-theoretically, if we want to detect the presence of each edge. In this paper, we assume $b_{\min}$ and the sparsity $d$ are known to the learner. We discuss extending our result to the case when $b_{\min}$ and $d$ are unknown in the Appendix E.3. More specifically, we can perform the full Algorithm 1 if $b_{\min}$ is unknown but the guarantee alters. We can also perform the first phase of our Algorithm 1 if $d$ is unknown.

We also define for a graph $G$ the following quantity $\tau(G)$, which is the maximum sum of squares of the coefficients for the out-edges of a node.

$$\tau = \tau(G) := 1 + \max_j \sum_{l:j \to l} b_{jl}^2 \tag{2}$$

We will see in the sequel how $\tau(G)$ arises naturally in both the upper and lower bounds on the sample complexity of the problem.

## 2. Results

In Section 2.1 we present an algorithm based on neighborhood selection and analyze its sample complexity. In Section 2.2 we present an information theoretic lower bound that matches this sample complexity up to a factor of $d$. Finally, in Section 2.3 we provide a polynomial time algorithm with slightly worse sample complexity.

## 2.1. Information Theoretic Recovery

The algorithm is split into two phases, as shown below.

---

**Algorithm 1** Estimating the DAG Information Theoretically

---
**input** Samples $(X^{(1)}, X^{(2)}, \ldots, X^{(m)})$
**output** Topology $\hat{G}$
 1: **Phase 1: Finding the Topological Ordering**
 2: $T = []$
 3: Find all $X_i$ with $\widehat{\text{Var}}(X_i) \leq (1 + b_{\min}^2/2)$ and put $i \in T$
 4: **while** $|T| < n$ **do**
 5:     $M = \emptyset$
 6:     **for** $i \in [n] \backslash T$ **do**
 7:         **for** $J \subseteq T, |J| = d$ **do**
 8:             $\beta_J \leftarrow$ Regress $X_i$ on $X_J$
 9:             $MSE \leftarrow \frac{1}{m} \sum_j \left( X_i^{(j)} - X_J^{(j)} \beta_J \right)^2$
10:             **if** $MSE \leq 1 + b_{\min}^2/2$ **then**
11:                 $T \leftarrow T.\text{append}(i)$
12:             **end if**
13:         **end for**
14:     **end for**
15: **end while**
16: **Phase 2: Finding the parents**
17: **for** $i \in T$ **do**
18:     $S \leftarrow$ nodes in $T$ before $i$
19:     **for** $J \subseteq S, |J| = \min(|S|, d)$ **do**
20:         PASSED $\leftarrow True$
21:         **for** $K \in S \backslash J, |K| \leq \min(|S| - |J|, d)$ **do**
22:             $\beta_{J \cup K} \leftarrow$ Regress $X_i$ on $X_{J \cup K}$
23:             **if** $\exists k \in K : |\hat{\beta}_k| \geq b_{\min}/2$ **then**
24:                 PASSED $\leftarrow False$, **break.**
25:             **end if**
26:         **end for**
27:         **if** PASSED $= True$ **then**
28:             $\hat{\beta} \leftarrow$ regress $X_i$ on $X_J$
29:             $\text{pa}(i) \leftarrow \{ j \in J : |\hat{\beta}_j| \geq \frac{b_{\min}}{2} \}$ , **break.**
30:         **end if**
31:     **end for**
32: **end for**

---

In Phase 1, we aim to find the topological order. A fundamental property of the equal-variances model is that the variance increases along directed paths. Thus, the strategy is to start with the node of smallest variance. Intuitively, we want to keep adding the node with the smallest conditional variance, conditioned on the nodes we have already added. One approach for computing this variance is by using the conditional variance formula for Gaussians, as is done in prior work (Chen et al., 2019; Gao et al., 2022). One important challenge in our setting though is that the expression for this conditional variance involves covariance matrices with arbitrarily bad condition numbers. Prior analyses in (Chen et al., 2019; Gao et al., 2022) bound the estimation

error for these matrices, which leads to a sample complexity that depends on a condition number assumption. Instead, in our analysis we take advantage of the relation between this conditional variance and the mean squared error of regressing the node with the previous nodes in the ordering, which results in significant gains in sample complexity.

Thus, for Phase 1, we still calculate the conditional variances as in (Gao et al., 2022) and (Chen et al., 2019). We observe that for any node $i$ and any set $J$, if $J$ contains all the parents of $i$, then $\text{Var}[X_i | X_J] = 1$. Otherwise, the conditional variance is larger than $1 + b_{\min}^2$. Thus, there is a gap between these two cases (see Proposition A.1) which helps us distinguish between them.

Phase 1 of the algorithm outputs an a list $T$, a valid topological order of all the nodes. In line 2, we initialize $T$ as an empty list. We know that the first node in the topological order has no parent, thus it has a variance of 1, the smallest variance among all random variables. Therefore, in line 3, we choose $i$ such that $X_i$ has a small empirical variance to be the first in the list.

Let us now discuss lines 4 to 15 of Algorithm 1. Suppose, inductively, that we have identified the correct $T$ for each iteration. In that case, all the nodes in $T$ appear earlier in the topological order than $V \backslash T$. This means that in the next iteration, the node that comes right after $T$ in the topological order has all its parents in $T$. We call that node $i$. Therefore, the conditional variance, $\text{Var}[X_i | X_T]$, is 1 (see Proposition A.1), and we can choose the one with smallest conditional variance accordingly.

However, we cannot directly calculate the conditional variance on $X_T$ or even estimate it with bounded accuracy by vanilla OLS. If we do so, we need $O(n)$ samples, which is larger than our target $O(\log n)$ samples, to estimate the conditional variance of $X_i$ given $X_T$. Nevertheless, we can take advantage of the sparsity of the graph, i.e. $|\text{pa}(i)| \leq d$. By conditional independence, we have $\text{Var}[X_i | X_{\text{pa}(i)}] = \text{Var}[X_i | X_T] = 1$. Therefore, instead of calculating the conditional variance $\text{Var}[X_i | X_T]$, we can calculate $\text{Var}[X_i | X_J]$ for all $\leq d$-element subsets $J$ of $T$. Hence, lines 7 to 13 do the following: For all $i$, we loop through $J \subseteq T, |J| \leq d$ to find the subset that makes the empirical conditional variance smaller than $1 + b_{\min}^2/2$. If we find such $i$, we append to topological order. We repeat this procedure until there is no node outside $T$.

After finding a topological ordering in the first phase, the second phase aims to recover the parents of each node, in order to completely determine the topology. Here, we draw inspiration from a technique developed in (Misra et al., 2020) for undirected Graphical Models. For any vertex $i$, we would like to figure out which of the candidate subsets of size at most $d$ that come before it is the correct parent set.

The only guarantee we have in this sample-starved setting is that linear regression with $O(d)$ covariates has small error. Thus, for any such candidate parent set $A$, we perform a series of regressions of $i$ with covariate set $A \cup B$, where $B$ ranges over all subsets of $d$ nodes before $i$ again. If for at least one of these sets $B$, some regression coefficient of a variable in $X_j \in B$ comes out sufficiently large, then we know that $A$ does not contain all parents and hence we reject it. The reason is that conditioned on the parent set, $X_i$ should be independent of all other variables that come before it in the ordering. On the other hand, if for all sets $B$, the only large coefficients come from variables in set $A$, then we know that $A$ contains all parents of $i$. We then pick the parents of $i$ as the variables of $A$ with large coefficients. A detailed description of this procedure is given in Algorithm 1. We remark that, for the second phase of finding the parents, it is possible to use as an alternative the Best Subset Selection algorithm. Even though the authors in (Gao et al., 2023) do not consider the equal variances model, it is possible to use their analysis about support recovery together with some additional steps to obtain similar sample complexity guarantees for the second phase of finding the parents.

The next result provides an upper bound on the number of samples required by Algorithm 1 to find the correct topology with high probability (proof: A.) Note that Algorithm 1 requires knowledge of the parameters $\sigma, d, b_{\min}$. In Supplementary E.3, we present an adaptive version of this algorithm with similar sample complexity guarantees.

**Theorem 2.1.** *Suppose we run Algorithm 1 using $m$ independent samples generated from a DAG $G$ according to* (17). *If Assumption 1.1 holds, then there exists an absolute constant $C > 0$, such that the following guarantees hold.*

- *Phase 1 of Algorithm 1 succeeds in finding a correct topological ordering of the nodes with probability at least $1 - \delta$, provided that*

$$m \geq C \frac{1}{b_{\min}^4} \left( d \log(n/d) + \log(1/\delta) \right)$$

- *Provided that Phase 1 of Algorithm 1 succeeds in finding a correct ordering, Phase 2 finds the correct parent set for each node with probability at least $1 - \delta$, provided that*

$$m \geq C \frac{\tau(G)}{b_{\min}^2} \left( d \log(n/d) + \log(1/\delta) \right)$$

Notice that we obtain separate characterizations for the samples required to find the correct topological ordering and the correct parent sets. We remark that for the task of finding the ordering, it suffices to assume that the noise is subgaussian (for details see Supplementary E.4). The only

relevant quantities for finding the topology are $n, d, b_{\min}$, with no dependence on $\kappa$. An additional relevant quantity for the sample complexity of determining the parent set is $\tau(G)$. We note here that this is always better than a condition number bound, as the following Lemma suggests.

**Lemma 2.2.** *For any probabilistic model of the form* (17) *with DAG $G$, we have $\tau(G) \leq \kappa$.*

The proof of this lemma can be seen in Appendix D. Thus, our bound is stronger than the ones obtained using condition number assumptions (Chen et al., 2019; Gao et al., 2022). Furthermore, there are also cases where $\tau(G)$ is a constant and $\kappa$ grows exponentially with $n$. For example, if we have the DAG $X_1 \to X_2 \to \cdots \to X_n$ and $b_{i,i+1} = k > 1$, then the condition number is at least $k^{n-1}$ but $\tau(G)$ is constant. A related construction involving trees with similar guarantees is given in the proof of Lemma 2.5.

For the bound $\tau(G)$, in Section 2.2, we show that this dependence on $\tau$ is necessary. For more detailed discussion and comparisons of this result with that of prior work, we refer the reader to Section 2.4.

## 2.2. Information Theoretic Lower Bound

In Section 2.1 we presented an algorithm that succeeds with high probability as long as it is given $O(\max(b_{\min}^{-4}, \tau b_{\min}^{-2}) d \log(n/d))$ samples. We now present a lower bound on the sample complexity of any algorithm, which matches the upper bound up to a factor of $d$. Thus, we obtain a fine-grained characterization of the optimal sample complexity for this problem.

Our general strategy will be to define a family of distributions as in (17), each indexed by a different topology and then apply Fano's inequality (Yu, 1997). Each family will instantiate some part of the difficulty of the problem, which corresponds to some term in the sample complexity. Concretely, we can prove (in Appendix B) the following.

**Theorem 2.3.** *We can construct a family of DAGs $\mathcal{G}$ satisfying Assumption 1.1 with the following property: suppose one DAG $G \in \mathcal{G}$ is picked uniformly at random from $\mathcal{G}$ and $m$ independent samples $X^{(1)}, \ldots, X^{(m)}$ are generated according to $G$. Then, any estimator $e : \mathbb{R}^{mn} \to \mathcal{G}$ that takes as input these samples satisfies*

$$\mathbb{P}\left[ e\left( X^{(1)}, \ldots, X^{(m)} \right) \neq G \right] \geq \frac{1}{2}$$

*as long as $m \leq C \max(b_{\min}^{-4}, \tau b_{\min}^{-2}) \log(n)$, where $C > 0$ is an absolute constant.*

Thus, Algorithm 1 is minimax optimal up to a factor of $d$. Each of the two terms in the max reflects the difficulty of finding the correct topology and finding the parent set,

respectively. Hence, we essentially characterize the complexity of these two problems separately up to a $d$-factor. This is explained in more detail in the Supplementary Material. For a more detailed comparison of this result with prior work see Section 2.4.

### 2.3. Efficient Algorithm with Bounded Variance

The runtime of Algorithm 1 is $O(n^{2d+2})$, which even for small values of the maximum degree becomes impractical. We are thus in search of a computationally efficient alternative, which will also enjoy favorable sample complexity guarantees. One important observation about structure learning is its similarity with Sparse Linear Regression (SLR), where one is tasked with estimating the regression vector in a linear regression problem, assuming at most $d$ of its $n$ entries are non-zero. Information theoretically, $O(d \log n)$ samples are sufficient to estimate the vector. However, it is conjectured that no polynomial time algorithm can succeed under general designs using $o(n)$ samples (Kelner et al., 2024). The connection with our problem comes from the fact that finding the parents of each node basically involves solving a SLR problem where the non-zero coefficients correspond to the parents of the node. Given this close connection, it seems unlikely that a polynomial time algorithm succeeds in finding the DAG using $o(n)$ samples, at least in the general case.

We thus need to pose extra assumptions in order to make the problem tractable. One common assumption, which is also present in prior work (Ghoshal & Honorio, 2017b), is that all individual variances are bounded by a constant.

**Assumption 2.4.** For all $i$, $\mathrm{Var}\left(X_i\right) \leq R^2$.

This can be seen as a mild assumption, as it rules out pathological cases where the variance of some nodes grows with the size of the DAG, a situation that could be considered unrealistic in most applications. It is also worth noting that this assumption is strictly weaker than a condition number bound. In particular, we can prove that a condition number bound implies a bound on the maximum variance, but the converse is not necessarily true (proof in Appendix D.2).

**Lemma 2.5.** *(i) Suppose we have variables $X_1, \ldots, X_n$ generated according to (17) and let $\kappa$ be the condition number of their covariance matrix. Then $\mathrm{Var}\left(X_i\right) \leq \kappa$ for all $i$.*

*(ii) For every $\alpha \in (0,1)$, there exists a sequence of DAGs $\{G_n\}_n$ indexed by the number of nodes, with maximum variance $R_n$ and condition number $\kappa_n$, such that $R_n = O(1)$ but $\kappa_n = \Omega(n^{\alpha})$.*

Our strategy for finding the DAG is to use Lasso to discover the neighborhood of each graph. We use the mean squared

error of the Lasso to determine the next node in the ordering. Afterwards, to find the parents of each node, we regress on nodes that appear before it in the topological ordering. The details appear in Algorithm 2, where $\lambda$ is the regularization parameter.

---

**Algorithm 2** Efficient algorithm for recovering the topology

---

**input** Samples $(X^{(1)}, X^{(2)}, \ldots, X^{(m)})$
**output** Topology $\hat{G}$
1: $T \leftarrow \left[i : \widehat{\mathrm{Var}}(X_i) \leq (1 + b_{\min}^2/2)\right]$
2: **while** $|T| < n$ **do**
3:     $M = \emptyset$
4:     **for** $i \in [n] \backslash T$ **do**
5:         Choose $\lambda = O(\sqrt{\log(n/\delta)} R/\sqrt{m})$
6:         $\hat{\beta} \leftarrow$ LASSO regression of $X_i$ on $X_T$
7:         $MSE \leftarrow \frac{1}{m} \sum_j \left(X_i^{(j)} - X_T^{(j)} \hat{\beta}\right)^2$
8:         **if** $MSE \leq 1 + b_{\min}^2/2$ **then**
9:             $M \leftarrow M \cup \{i\}$
10:           $\mathrm{pa}(i) \leftarrow \{j \in J : |\hat{\beta}_j| \geq b_{\min}/2\}$
11:         **end if**
12:     **end for**
13:     $T \leftarrow T$.append$(M)$
14: **end while**

---

Below we present an upper bound on the sample complexity of Algorithm 2 (proof is in Appendix C.)

**Theorem 2.6.** *Under Assumptions 1.1 and 2.4, there is an absolute constant $C$ such that Algorithm 2 run with $m$ samples recovers the correct DAG with probability at least $1 - \delta$, as long as*

$$m \geq C(R^2 \cdot \log(n/\delta) \cdot d^4 \cdot \tau^3 \cdot b_{\min}^{-4})$$

*Moreover, Algorithm 2 runs in time $poly(n, m, \tau, R)$.*

A comparison with prior work follows in the next Section.

### 2.4. Discussion and Comparison

For completeness of presentation, we now offer a more detailed technical comparison of our results with the most relevant prior work.

Regarding upper bounds on the sample complexity, (Ghoshal & Honorio, 2017b) obtain a sample complexity of $O(k^4 \log n)$, where $k$ is the size of the maximum Markov blanket, which could be much larger than the maximum in-degree. Furthermore, they need to assume that the infinity norm of $\Theta$ is bounded, which is a form of condition number assumption that we do not require. In (Chen et al., 2019) they obtain an improved $O(d^2 \log n)$ dependence, but they also need an upper bound on the maximum variance of each node in the graph. Finally, (Gao et al., 2022) introduce an algorithm that needs $O(M^5 d \log n)$ samples, which

has the optimal dependence on $d$ and $n$, but also depends on $M$, which is an upper bound on the square root of the condition number of $\kappa$ of $\Sigma$. In contrast, Algorithm 1 needs $O(\max(b_{\min}^{-4}, \tau b_{\min}^{-2}) d \log(n/d))$ samples, which is also optimal in the dependence on $n$ and $d$, but only depends on $\tau$, which is smaller than $\kappa$ by Lemma 2.5.

The quantity $\tau$ is information theoretically necessary, as Theorem 2.3 implies an $\Omega(\max(b_{\min}^{-4}, \tau b_{\min}^{-2}) \log(n))$ lower bound. The only lower bound in prior work that is directly comparable to ours is the one in (Gao et al., 2022). Using our notation, Theorem 3.1 from their work establishes the lower bound $\Omega(\max(\frac{d \log(n/d)}{M^2-1}, b_{\min}^{-2} \log n))$. It is clear that our bound has the improved $b_{\min}^{-4}$ factor compared to $b_{\min}^{-2}$ of (Gao et al., 2022). At first glance, it seems that their bound $\Omega(\frac{d \log(n/d)}{M^2-1})$ is stronger by a $d$ factor. However, we can show (see Appendix D.6.) that $M^2 - 1 \geq d b_{\min}^2$ always holds, which means that the bound in (Gao et al., 2022) reduces to $\frac{\log n}{b_{\min}^2}$, which is strictly worse than the lower bound in Theorem 2.3. In fact, this $d$-factor discrepancy between upper and lower bounds is a known limitation in the literature on learning undirected GGMs (Wang et al., 2010; Misra et al., 2020; Kelner et al., 2020). Thus, improving the lower bound for our problem could have significant implications for other related questions as well. Also, (Ghoshal & Honorio, 2017a) establishes general lower bounds for Bayesian Networks where the conditional distributions are exponential families. However, when instantiated in the equal variances model, these results scale as $\frac{k \log n}{w_{\max}^2}$, where $w_{\max}$ is the maximum $\mathcal{L}_2$-norm of the coefficients vector of the parents of a node. Thus, this bound is of the order of $\log n$ and fails to capture the dependence on $\tau$.

Our result also gives us the opportunity to compare the problems of learning directed and undirected graphs in relatively equal footing. In (Misra et al., 2020), they propose an algorithm that uses $(d/\beta_{\min}) \log n$ samples to find the undirected graph, where $\beta_{\min}$ is the minimum normalized edge strength. Thus, we can view $\tau$ as the analog of $\beta_{\min}$ for directed graphs.

Regarding the efficient algorithm, the most relevant prior work to compare with is (Ghoshal & Honorio, 2018). They perform $\mathcal{L}_1$-regularized Gaussian MLE and their analysis assumes that the inverse covariance matrix of the variables is upper bounded in $\mathcal{L}_1$-norm. This assumption is qualitatively different from the ones we make, which makes the two results incomparable.

In addition, Algorithm 1 needs knowledge of $d, b_{\min}$ to perform. We give a modification of the algorithm when $\sigma^2$, $d$ or $b_{\min}$ are unknown to the learner, and we refer the reader to Appendix E.3, where a similar guarantee (for Phase 2) is obtained with a similar order of sample complexity.

We also note that Algorithm 1 is not directly comparable

to the PC algorithm (Spirtes et al., 2001). Although PC algorithm does not require the structural equation as Equation (17), the faithfulness assumption is not always satisfied because of the path cancellation. We also performed experimental simulation for the PC algorithm, and the result shows that the PC algorithm behaves worse than Algorithms 1, 2, and (Gao et al., 2022) even for the case of small $n, d$. Further comparison and discussion can be found in Section E.2 in the supplementary material.

Finally, Phase 1 of Algorithm 1 can be extended to the case when $\varepsilon_i$ are only centered sub-Gaussian with equal variance (see Section E.4).

## 3. Proof Sketch of the Analysis

Before presenting the proof sketch for Theorem 1, we would like to highlight the technical novelty that leads to an algorithm with sample complexity that is independent of the condition number. The general technique is to use the gap between conditional variances if some parents are missing from the conditioning set. For Algorithm 1, we conduct a more detailed analysis of the conditional variance to achieve a better sample complexity. Specifically, we use concentration properties of the OLS error for both Phase 1 and Phase 2 in Algorithm 1. For Phase 1, the analysis of the conditional variance is sufficient to make the claim in Theorem 1, which is a bound already free of conditional number. However, for Phase 2, the variance gap only leads to $O(d \log(n) \cdot \frac{\tau(G)^2}{\beta_{\min}^4})$ sample complexity, where the ratio $\frac{\tau(G)^2}{\beta_{\min}^4}$ is not information optimal. Therefore, we shift the method to additionally examine the OLS coefficients in (Misra et al., 2020) to distinguish the correct parent set. Therefore, we can then sharpen the sample complexity to $O(\log(n)/d \cdot \frac{\tau(G)}{\beta_{\min}^2})$ by using this method. For Algorithm 2, we employ LASSO regression and also utilize the technique of examining the MSE and the coefficients. For the MSE part (similar to Phase 1 in Algorithm 1), we can reuse the proof in Theorem 2.1, and we replace the analysis for Phase 2 in Algorithm 1 with LASSO results for the coefficients in (Wainwright, 2019).

We now provide a proof sketch for Theorem 2.1 (for Algorithm 1) and Theorem 2.6 (for Algorithm 2), which can be analyzed using the same basic principles. Suppose we have a variable $X_i$ and a subset $I \subseteq [n] \setminus \{i\}$ with $|I| \leq d$. Since our algorithms operate inductively, we will always assume that all nodes in $I$ come before $i$ in the topological ordering. The basic observation comes from analyzing the regression

$$\hat{\beta} := \underset{\beta}{\operatorname{argmin}} \frac{1}{m} \sum_{j=1}^{m} \left( X_i^{(j)} - \beta^\top X_I^{(j)} \right)^2$$

$$\beta^* := \underset{\beta}{\operatorname{argmin}} \mathbb{E} \left[ \left( X_i^{(j)} - \beta^\top X_I^{(j)} \right)^2 \right]$$

In particular, we care about the empirical and population *Mean Squared Error* (MSE) of this regression, which is

$$\widehat{MSE}(X_i, X_I) = \frac{1}{m} \sum_{j=1}^{m} \left( X_i^{(j)} - \hat{\beta}^\top X_I^{(j)} \right)^2$$

$$MSE(X_i, X_I) = \mathbb{E} \left[ \left( X_i^{(j)} - (\beta^*)^\top X_I^{(j)} \right)^2 \right]$$

Since all the variables are Gaussian, we can also write

$$MSE(X_i, X_I) = \mathrm{Var}\left(X_i | X_I\right)$$

In order to estimate the conditional variance $\mathrm{Var}\left(X_i|X_I\right)$, previous analyses ((Chen et al., 2019; Gao et al., 2022)) use the well known identity

$$\mathrm{Var}\left(X_i|X_I\right) = \Sigma_{ii} - \Sigma_{iI}\Sigma_{II}^{-1}\Sigma_{Ii},$$

where we use the indices to denote subsets of the rows and columns. They proceed by estimating each of these subblocks of the covariance matrix, which requires condition number assumptions. Instead, we use the observation that this conditional variance is equal to the MSE of the regression of $X_i$ with $X_I$. Using simple properties of our model, we can establish that $\widehat{MSE}(X_i, X_I)$ will be close to $MSE(X_i, X_I)$ when using $O(d \log n)$ samples, with no dependence on the condition number.

Continuing with the proof, we distinguish between two possible cases:

- If $\mathrm{pa}(i) \subseteq I$, then clearly $\mathrm{Var}\left(X_i|X_I\right) = \mathrm{Var}\left(X_i|X_{\mathrm{pa}(i)}\right) = \mathrm{Var}(\varepsilon_i) = 1$, by conditional independence of $X_i$ from all previous nodes, given the parents.

- If $\mathrm{pa}(i) \setminus I \neq \emptyset$, meaning that some parents are omitted from $I$, then $\mathrm{Var}\left(X_i|X_{\mathrm{pa}(i)}\right) = \mathrm{Var}(\epsilon_i) + \mathrm{Var}\left(\sum_{j \in \mathrm{pa}(i)} b_{ji}X_j|X_I\right) > 1$.

Thus, our ability to distinguish between these two cases depends on how big the variance gap between them is. A smaller gap would make the task harder. We now explain why the tasks of finding the ordering and finding the parents have *different* variance gaps, which will result in different factors appearing in the sample complexity of each task.

### 3.1. Phase 1: Finding the topological ordering

Algorithm 1 works inductively. Suppose we have found the first $r$ nodes in the ordering, which is the subset $T_r$. For every $i \notin T_r$, $i$ should be the next node in the ordering only if all of its parents lie in $T_r$. If it does, then there exists a subset $I \subseteq T_r$ with $|I| = d$ such that $\mathrm{Var}(X_i|X_I) = 1$. On the other hand, if there exists $j \notin T_r$ such that

$j \to i$, then for every $I \subseteq T_r$ we have $\mathrm{Var}(X_i|X_I) \geq \mathrm{Var}(b_{ji}\epsilon_j + \epsilon_i|X_I) = b_{ij}^2 + 1 \geq 1 + b_{\min}^2$, where we used the fact that $j$ comes after $I$ in the ordering. Thus, we can see that the variance difference between the two cases is at least $b_{\min}^2$. The empirical variance behaves like a chi-squared distribution, which is subexponential. Thus, in order to distinguish between these two cases, we need $O(b_{\min}^{-4})$ samples. This is how the first bound of Theorem 2.1 arises.

### 3.2. Phase 2: Finding the parents

After we find the ordering, we turn to locate the parents for each node $i$. Let $B_i$ be the nodes before $i$ in the ordering. Suppose a subset $I \subseteq B_i$ contains all the parents of $i$. Then $\mathrm{Var}(X_i|X_I) = 1$. Now suppose there exists $j \in \mathrm{pa}(i)$ with $j \notin I$. As we observed earlier, in that case $\mathrm{Var}(X_i|X_I) \geq 1 + \mathrm{Var}(b_{ji}X_j|X_I) = 1 + b_{ji}^2\mathrm{Var}(X_j|X_I)$. The situation now is different because $I$ could also contain some descendants of $j$. In general, we establish the following Lemma.

**Lemma 3.1.** *For any $i \in [n]$ and any $I \subseteq [n] \setminus \{i\}$, we have $\mathrm{Var}(X_i|X_I) \geq 1 + b_{\min}^2/\tau$.*

Thus, $\tau$ arises exactly as a lower bound in the conditional variance of a node if we also condition on some of its descendants. From Lemma 3.1 it becomes clear that the variance gap is now lower bounded by $b_{\min}^2/\tau$. Thus, by implementing this strategy of finding the parents, we would need $O(\tau^2/b_{\min}^4)$ samples to distinguish the true parent set, but it does not give the optimal dependence on $\tau$ or $b_{\min}$.

To obtain the optimal dependence, we instead use a strategy that was employed in (Misra et al., 2020) to learn undirected GGMs. Specifically, instead of using the MSE to distinguish the correct parent set, we use the coefficients computed by the regression. The idea is that if a subset $I$ contains all the parents, then if we regress $X_i$ with $X_{I \cup J}$ for any other $J \subseteq B_i$, the coefficients of the regression vector in $J$ will be small. On the other hand, if there exists $j \in \mathrm{pa}(i)$ with $j \notin I$, then there exists a set $J$ with $j \in I$, such that $I \cup J$ includes all parents. Thus, if we regress $X_i$ with $X_{I \cup J}$, the coefficient for $X_j$ will be large. So, this becomes our distinguishing criterion for the correct neighborhood. It turns out that if we have $O(\tau/b_{\min}^2)$ samples, the accuracy with which we can compute the coefficients in the regression is sufficient to make that distinction. Hence, we obtain the optimal sample complexity.

## 4. Simulation Results

To validate our findings, we run Algorithm 1, Algorithm 2, and the algorithm in (Gao et al., 2022) on synthetic datasets generated by DAGs) and compare their performance. The samples are generated i.i.d. from a randomly generated DAG $G$, which is constructed as follows: (1) Draw a random

permutation of $\{0, 1, \ldots, n-1\}$ as $\sigma(0), \ldots, \sigma(n-1)$.

(2) For each node $\sigma(j)$, choose $\sigma(0), \ldots, \sigma(j-1)$ as a parent of $\sigma(j)$ i.i.d. with probability $\min(\frac{d}{j+2}, \frac{1}{2})$. Then, truncate the parent set to at most $d$ nodes.

(3) For each edge $i \rightarrow j$, we assign $b_{i,j}$ i.i.d. uniformly sampled from $[-b_{\max}, -b_{\min}] \cup [b_{\min}, b_{\max}]$ where $0 < b_{\min} < b_{\max}$ are tunable parameters.

We note that this is similar to the topology that was chosen in (Gao et al., 2022) to simulate their algorithm. After sampling the graph, we generate $m$ independent samples according to the model specified by equation (17).

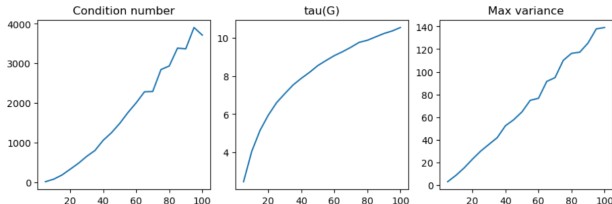

*Figure 1.* Growth of condition number, $\tau$ and maximum variance as $n$ grows, averaged over 1000 random graphs with same $d = 4, b_{\min} = 0.5, b_{\max} = 1$.

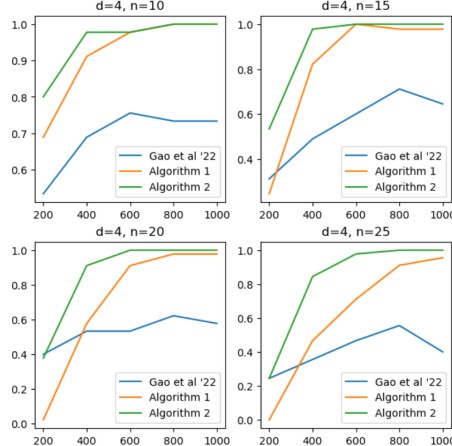

*Figure 2.* Accuracy results for three algorithms. The blue lines represent the algorithm in (Gao et al., 2022), the orange line is Alg. 1, and the green line is Alg. 2. The $x$-axis represents the number of samples $m$, and the $y$-axis represents the accuracy.

Figure 2 shows the results of running the three Algorithms with an increasing number of nodes $n = 10, 15, 20, 25$, degree $d = 4$, $b_{\min} = 0.5$, $b_{\max} = 1$, and an increasing number of samples $m = 200, 400, 600, 800, 1000$. For each combination of parameters $n, d, m$, the accuracy is tested by generating 45 random graphs, running all three algorithms on them, and calculating the fraction of successful recoveries of the graph.

We also present a plot (Figure 3) of the average number of false positive (reported but not in the ground truth) edges for the graph with the 95% confidence intervals. We observe that the number of false positive edges in Algorithms 1, 2,

and (Gao et al., 2020) have a trend of decreasing to zero, and Algorithms 1, 2 converge faster then (Gao et al., 2020) when the number of samples exceeds 600. More simulations and code, including comparisons with the PC algorithm, can be found in Supplementary Materials E.2, F, and G.

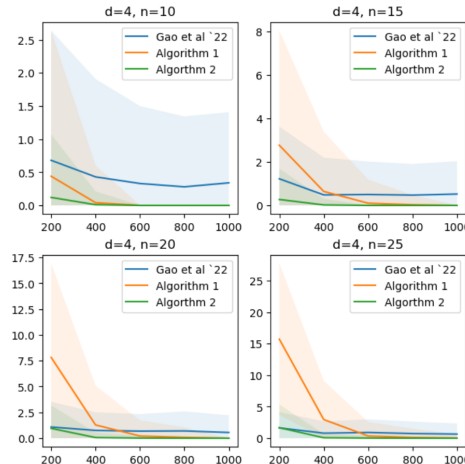

*Figure 3.* False positive results for three algorithms. We still use blue lines to represent the algorithm in (Gao et al., 2022), the orange line is Alg. 1, and the green line is Alg. 2. The $x$-axis represents the number of samples $m$, and the $y$-axis represents the average false positive edges. Also, the shade represents the 95% confidence region (centered with mean and $\pm 2\sigma$ error region) of each case.

The first thing to notice is that our algorithms seem to be less accurate than (Gao et al., 2022) for a small number of samples (up to around 400 samples). On the other hand, our methods clearly outperform (Gao et al., 2022) once the number of samples becomes large enough, with the latter even failing to be consistent. One possible explanation for this behavior is that the sample complexity of Algorithms 1 and 2 have a worse constant than (Gao et al., 2022). For Algorithm 1, in order to find the parents, we iterate over all $d$-sized subsets both to find the candidate neighborhood and to test it with respect to all other neighborhoods, while (Gao et al., 2022) simply choose the subset with the smallest conditional variance, which could result in a better constant. But if the number of samples becomes even slightly larger than 400, especially for larger $n$, the effect of the condition number begins to show, as (Gao et al., 2022) fails to converge to the truth. Indeed, in Figure 1, we have plotted the growth of the average condition number $\kappa$, $\tau(G)$, and maximum variance $R$ as $n$ increases. It is clear that $\kappa$ grows superlinearly, $\tau$ grows sublinearly, and $R$ grows linearly. This could explain the deteriorating performance of (Gao et al., 2022) when $\kappa$ increases, compared to our methods, which remain consistent.

The experiments were run using Python with `numpy` and `scikit-learn` package on an 11th Gen Intel Core i7-11800H 2.30 GHz CPU with 16GB of memory.

## Impact Statement

This paper presents work whose goal is to advance the field of Machine Learning. Specifically, this paper presents new algorithms for learning a specific type of Bayesian Network, a graphical model that occurs in many disciplines. Although these algorithms are novel, this paper contains theoretical results. We have discussed the related work and open directions, which mainly focus on the theoretical aspects of the problem. The data generated for the empirical experiments are synthetic and do not have any privacy issues. We do not exclude the possibility that there are potential societal consequences of our work, none of which we feel must be specifically highlighted here.

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

# A. Proof of Theorem 2.1

The Algorithm consists of two parts: finding the topological order and finding the parents. We use the notation $i \preceq j$ ( $i \prec j$ ) to denote that $i$ comes (strictly) before $j$ in the topological ordering. $\succeq$ and $\succ$ are defined similarly.

We will sometimes use the notation $a \leftarrow b$ when we want to apply some Lemma that has variable $b$ with the value $a$. e.g. in the proof of Theorem 2.1, we use Lemma A.3, we are going to take $\delta$ in the Lemma A.3 as $\delta/(2N)$, where this $\delta$ is the $\delta$ in Theorem 2.1

As we have explained in Section 3, the general approach in both phases of the algorithm is to use the Mean Squared Error (MSE) of linear regression to estimate the conditional variance of a node given some subset of nodes that appear before it in the ordering. The variance gap between correct and incorrect subsets of nodes will be different in each phase, which gives rise to different sample complexities for finding the ordering and finding the parents. We give the details for each phase below. Without loss of generality, $n \geq 2$, $b_{\min} \leq \frac{1}{\sqrt{3}}$ in the above discussion.

## A.1. Finding the ordering

Algorithm 1 builds the topological ordering iteratively by finding at each step the next node in the topological order. Suppose inductively that the first $t$ nodes form a valid topological ordering, for some $t < n$ and let $T$ be this subset of nodes. We will show that Algorithm 1 correctly identifies the $t + 1$-th node in the ordering. To achieve that, it considers all other nodes $i \in [n] \setminus T$ and calculates the conditional variance of $i$ given subsets $J \subseteq T$ with $|J| \leq d$. This is the step where previous analyses ((Chen et al., 2019; Gao et al., 2022) introduce condition number assumptions, in order to estimate this conditional variance. Instead, we will use the MSE of linear regression to estimate this conditional variance. Indeed, we can show that the empirical and true MSE are close to each other using a number of samples that is independent of the condition number.

The Empirical mean squared error consists of two parts. We can write it as

$$
\widehat{MSE} = \frac{1}{m} \sum_{j=1}^{m} (X_i^{(j)} - X_J^{(j)} \beta_J)^2
$$

$$
= \underbrace{\frac{1}{m} \sum_j (X_i^{(j)} - X_J^{(j)} \beta_J^*)^2}_{\text{variance error}} + \underbrace{\frac{1}{m} \sum_j (X_J^{(j)} \hat{\beta}_J - X_J^{(j)} \beta_J^*)^2}_{\text{beta error}} + 2 \frac{1}{m} \sum_j (X_i^{(j)} - X_J^{(j)} \beta_J^*)(X_J^{(j)} \hat{\beta}_T - X_T^{(j)} \beta_J^*)
$$

In the above, we have identified the two key quantities of the error, which we call the Emprirical Variance Error (call it $VE$) and Empirical Beta Error (call it $BE$) accordingly. By Cauchy-Schwartz inequality, we have

$$
VE^2 + BE^2 - 2\sqrt{VE \cdot BE} \leq \widehat{MSE} \leq VE^2 + BE^2 + 2\sqrt{VE \cdot BE}
$$

In order to decide whether $i$ can be the next node in the ordering, we have to determine whether all parents of $i$ are contained in $T$ or not. The following Lemma shows that, in the population case (infinite samples) there is way to distinguish between these two cases by looking at the conditional variance.

**Proposition A.1.** *(1) If $T$ contains all the parents, then there exists some $J \subseteq T$ and $|J| \leq d$ such that the conditional variance of $X_i|x_J$ is 1.*

*(2) If $T$ does not contain all the parents, then there exists a parent that is not in $T$, so for any coefficient vector $\beta_T$ each $X_i - X_T \beta_T$ is a Gaussian random variable that has variance at least $1 + b_{\min}^2$. Specifically, for any $J \subseteq T$, for any coefficient $\beta_J$, $X_i - X_J \beta_J$ also has variance $\geq 1 + b_{\min}^2$.*

*Proof.* This property is also proved in (Gao et al., 2022) , but for completeness we provide a proof. We first prove Property A.1. The first point follows immediately from the definition. For the second point, let $k$ be the last parent of $i$ in the topological order. By the preceding discussion, it follows that $k \notin T$. Let $T_k$ be the set of nodes that is prior to $k$. So

$T \subseteq T_k$ Then, we have

$$\text{Var}(X_i - x_T \beta_T) = \mathbb{E}_{X_{T_k}}[\text{Var}(X_i - X_{T_k}\beta_T | X_{T_k})] + \text{Var}[\mathbb{E}(X_i - X_T \beta_T) | X_{T_k}] \tag{3}$$

$$\geq \mathbb{E}_{X_{T_k}}[\text{Var}(X_i - X_T \beta_T | X_{T_k})] = \mathbb{E}_{X_{T_k}}[\text{Var}(X_i | X_{T_k})] \tag{4}$$

$$= \mathbb{E}_{X_{T_k}}[\mathbb{E}_{X_k}[\text{Var}(X_i | X_{T_k \cup \{k\}})]] + \mathbb{E}_{X_{T_k}}[\text{Var}[\mathbb{E}_{X_k}(X_i | X_k) | X_{T_k}]] \tag{5}$$

$$= 1 + \mathbb{E}_{X_{T_k}}[\text{Var}(b_{k \to i} X_k) | X_{T_k}] = 1 + b_{k \to i}^2 \geq 1 + b_{\min}^2 \tag{6}$$

We explain each line: (4) is by the law of total variance, (5) is because the variance is greater then zero and $X_T \beta_T$ is a constant under the condition of $X_{T_k}$ because $T \subseteq T_k$, (6) is again by law of total variance, (7) is because while conditioning on $X_{T_k}$ and $X_k$, all parents of $i$ are conditioned. Thus the variance of $X_i$ is 1 because of the model. Also, the expected value of $X_i$ condition on $X_{T_k}$ and $X_k$ is $X_i = \sum_j b_{j \to i} X_j = \sum_{j \neq k} b_{j \to i} X_j + b_{k \to i} X_k$. Since conditioning on $X_{T_k}$, that is all prior nodes than $k$, the value $\sum_{j \neq k} b_{j \to i} X_j$ is a constant, so the conditional variance of $\mathbb{E}(X_i | X_k)$ is the same as that of $b_{k \to i} X_k$, which by the definition of the model is $b_{k \to i}^2$ times 1, which is $b_{k \to i}^2$. $\square$

The preceding property holds in the population setting. We would like to establish a similar separation using finite samples. This is done using the following Lemmas. The first one quantifies how large $BE$ is when the set $J$ might or might not contain all parents of node $i$. To state it, it will be helpful to we extend the meaning of the ground truth coefficient: since the joint distribution $(X_J, X_i)$ is a multivariate Gaussian distribution with covariance matrix $\begin{pmatrix} \Sigma_{JJ} & \Sigma_{Ji} \\ \Sigma_{iJ} & \Sigma_{ii} \end{pmatrix}$ we can always write

$$\mathbb{E}[X_i | X_J] = (\beta^*)^\top X_J$$

where $\beta^* = \Sigma_{JJ}^{-1}\Sigma_{Ji}$. We call this vector $\beta^*$ the **population coefficients** of $X_i$ with respect to $X_J$. The reason is that if we performed linear regression of $X_i$ on $X_J$ with infinite samples, the result would be $\beta_J^*$ and the conditional variance is $\Sigma_{ii} - \Sigma_{iJ}\Sigma_{JJ}^{-1}\Sigma_{Ji}$. If $J$ contains all of $i$'s parents and none of its descendants, then we have the population coefficiens of $X_i$ wrt $X_J$ are exaclty the coefficients $b_{j \to i}$.

**Lemma A.2.** *(For beta error.) In Algorithm 1, let $J \subseteq [n] \setminus \{i\}$ with $|J| = k \leq d$ and $\text{Var}(X_i | X_J) = V^2$, $\beta^*$ be the populations coefficients of $X_i$ w.r.t. $X_J$ and let $\hat{\beta}_J$ be the result of linear regression of $X_i$ on $X_J$ using $m$ independent samples $X^{(1)}, \ldots, X^{(m)}$ drawn from this distribution. Then, there is a universal constant $C_B = 4$, such that for any $\varepsilon < 1/2, \delta < 1/2$, if we are given $m \geq C_B(\log(1/\delta))/\varepsilon$ samples, then*

$$\frac{1}{m} \sum_{j=1}^{m} (X_T^{(j)}(\beta_J^* - \hat{\beta}_J))^2 \leq \varepsilon V^2$$

*with at least $1 - \delta$ probability.*

Thus, this shows that $BE$ will be relatively small if $J$ contains all parents. The next Lemma will be used to argue that $VE$ will also be small in that case. It's proof is standard and uses the subexponential property for the $\chi^2$ distribution.

**Lemma A.3.** *(For variance error.) Let $Y_1, Y_2, \ldots, Y_m$ be i.i.d. Gaussian random variables with variance $\sigma^2$. Then, there is a universal constant $C_V = 6$ such that for any $\varepsilon < 1/2, \delta < 1$, if we are given $m \geq C_V(\log(2/\delta)/\varepsilon^2)$ samples, then with at least $1 - \delta$ probability, we have*

$$(1 - \varepsilon)\sigma^2 \leq \frac{1}{m} \sum_{i=1}^{m} Y_m^2 \leq (1 + \varepsilon)\sigma^2$$

The proof of the Lemma A.2 is presented in Section D.3 and the proof for Lemma A.3 is in Section D.4. We are now ready to prove the first assertion of Theorem 2.1, which says that Phase 1 of Algorithm 1 correctly identifies the topology with high probability. Aided with these two lemma, we return to the proof of Theorem 2.1

*Proof.* Let's define $N = n(\binom{n}{0} + \binom{n}{1} + \cdots + \binom{n}{d})$, which is an upper bound of the number of linear regressions performed in Phase 1 of Algorithm 1. We would like to argue that all of these regressions succeed with high probability, so we employ

a union bound. This involves bounding $\log N$, which we now do. Since we have $n >> d$, we have $N < 2n\binom{n}{d}$ and thus by Stirling Formula,

$$\log N \leq \log 2 + (d+1)\log n - d\log(d/e) = 2 + (d+1)\log(n/d) + d + \log d < 5d\log(n/d).$$

Now let's assume inductively that Algorithm 1 has identified the first $t$ nodes in the ordering correclty and let $T$ be this subset of nodes. The Algorithm then examines every $i \in [n] \setminus T$ to determine if it is the next in the ordering. There are two possible cases.

$T$ **contains all the parents of** $i$. If $T$ contains all the parents of $i$, then $i$ can be used as the $t+1$-th node in the ordering. We prove that Algorithm 1 will indeed select it. By definition, there exists a $J \subseteq T$ with $|J| \leq d$ that contains all the parents. We want to upper bound the MSE of regressing $X_i$ on $X_J$. To apply A.3, we notice that $X_i^{(j)} - X_J^{(j)}\beta_J^*$ are i.i.d. $\mathcal{N}(0,1)$ random variables.

Suppose it holds that

$$VE \leq 1 + b_{\min}^2/8 \quad , \quad BE \leq b_{\min}^4/64. \tag{7}$$

Then, wherever $b_{\min} < 2$ (which we can wlog assume ) the empirical MSE can be bounded as

$$MSE \leq VE + BE + 2\sqrt{VE \cdot BE} \leq 1 + b_{\min}^2/8 + b_{\min}^4/64 + 2\sqrt{1 + b_{\min}^2/8} \cdot b_{\min}^2/8 < 1 + b_{\min}^2/2. \tag{8}$$

Thus, the condition in the If statement of line 9 of Algorithm 1 would be satisfied, which means $i$ would be selected as the next node. We now establish that (7) holds with high probability. We first notice that $\mathrm{Var}(X_i|X_J) = \mathrm{Var}(\epsilon_i) = 1$. Thus, we can apply Lemma A.2 by plugging in $\delta \leftarrow \delta/(2N)$, $\varepsilon = b_{\min}^4/64$. We also notice that $X_i^{(j)} - X_J^{(j)}\beta_J^*$ are i.i.d. $\mathcal{N}(0,1)$ random variables by definition of the model. Thus, we can apply Lemma A.3 by plugging in $\delta \leftarrow \delta/(2N)$, $\varepsilon = b_{\min}^2/8$. , By a union bound, these two lemmas give that with probability at least $1 - \delta/N$ , (7) holds for all regressions in Phase 1, as long as the number of samples satisfies

$$m \geq \max\left(C_B(k + \log(2N/\delta))/(b_{\min}^4/64), C_V \log(4N/\delta)/(b_{\min}^2/8)^2\right).$$

Therefore, if we choose $C = 5120 \geq \max(1280C_B, 640C_V)$, $m > C(d\log(n/d) + \log(1/\delta))/b_{\min}^4$ will imply the previous bound on $m$.

$T$ **does not contain all parents of** $i$ We want to establish that $i$ will not be selected as the next node. Let $J \subseteq T$ be any subset of $T$ that is considered by the algorithm for $i$. By Property A.1 we have

$$\mathrm{Var}(X_i|X_T) \geq 1 + b_{\min}^2,$$

Thus, the law of total variance yields

$$\mathrm{Var}(X_i|X_J) = \mathbb{E}_{X_{J\setminus T}}[\mathrm{Var}(X_i|X_T)] + \mathbb{E}[\mathrm{Var}(X_i|X_{T\setminus J})|X_J] \geq \mathbb{E}_{X_{J\setminus T}}[\mathrm{Var}(X_i|X_T)] \geq 1 + b_{\min}^2.$$

Letting $V^2 = \mathrm{Var}(X_i|X_J)$, our goal will be to lower bound the empirical $MSE$. Suppose the following holds

$$VE \geq (1 - b_{\min}^2/8)V^2 \quad , \quad BE \leq (b_{\min}^4/64)V^2, \tag{9}$$

then this would imply (the third inequality is due to the monotonicity of $VE$.)

$$\frac{MSE}{V^2} \geq VE + BE - 2\sqrt{VE \cdot BE} > VE - 2\sqrt{VE \cdot BE} \geq 1 - b_{\min}^2/8 - 2\sqrt{1 - b_{\min}^2/8} \cdot b_{\min}^2/8 > 1 - \frac{3}{8}b_{\min}^2.$$

Thus, since $V^2 > 1 + b_{\min}^2$, this would imply for $b_{\min} > \frac{1}{\sqrt{3}}$ (again, we can w.l.o.g. assume this),

$$MSE \geq (1 - \frac{3}{8}b_{\min}^2)V^2 > 1 + b_{\min}^2/2.$$

That would mean that the condition in the If statement of line 9 of Algorithm 1 is not satisfied, which means $i$ will not be selected as the next node. Establishing that (9) holds with probability at least $1 - \delta$ then involves a similar application of LemmasA.2 and A.3 to the one we saw in the previous case, which we omit.

$\square$

### A.2. Finding the parents

The next phase focuses on finding the parents of each node $i$, assuming we know the subset $T$ of nodes that come before it in the topological ordering. Here, we can make a connection between the linear regression problem for the directed graphs and the one for undirected ones. In particular, we focus on the marginal distribution of $(X_i, X_T)$, which is itself coming from a Bayesnet, which is obtained from the original Bayesnet by keeping all nodes up to $i$ in the ordering.

We start by proving some preliminary properties of the inverse covariance of a Bayesnet model in the following lemma:

**Lemma A.4.** *Let $(X_1, X_2, \ldots, X_n)$ be a multivariate Gaussian, with covariance matrix $\Sigma$ and inverse covariance matrix $\Theta = \Sigma^{-1}$, distributed according to model 17. Then we have: for $i \neq j$,*

$$\Theta_{ij} = b_{i \to j} + b_{j \to i} + \sum_l b_{i \to l} b_{j \to l},$$

*and for any $i$,*

$$\Theta_{ii} = 1 + \sum_j b_{i \to j}^2.$$

*Proof.* We can show it by calculating the likelihood of $X = (X_1, X_2, \ldots, X_n)$: it is $\exp(-\frac{1}{2} X^\top \Theta X)$ because of the joint Gaussianity. Let $\sigma(1), \sigma(2), \ldots, \sigma(n)$ be the topological order. By the Bayes rule, the likelihood of $X = (X_1, X_2, \ldots, X_n)$ is also

$$p(X_1, X_2, \ldots, X_n) = \prod_{j=1}^n p(X_{\sigma(i)} | X_{\sigma(1), \ldots, X_{\sigma(i-1)}})$$

$$= \prod_{j=1}^n \exp\left(-\frac{1}{2}(X_{\sigma(i)} - \sum_{j=1}^{i-1} b_{\sigma(i-1) \to \sigma(i)} X_{\sigma(i-1)})^2\right)$$

$$= \exp\left(-\frac{1}{2} \sum_{i=1}^n (X_{\sigma(i)} - \sum_{j=1}^{i-1} b_{\sigma(i-1) \to \sigma(i)} X_{\sigma(i-1)})^2\right)$$

$$= \exp\left(-\frac{1}{2} \sum_{i=1}^n (X_i - \sum_{j \to i} b_{j \to i} X_j)^2\right)$$

.

So, we have

$$X^\top \Theta X = \sum_{i=1}^n (X_i - \sum_{j \to i} b_{j \to i} X_j)^2.$$

Therefore, $\Theta_{ii}$ is the coefficient of $X_i^2$, in the above sum. We get a contribution of 1 from the term $(X_i - \sum_{j \to i} b_{j \to i} X_j)^2$, and a contribution of the sum of $b_{i \to j}^2$ from the sum of $(X_j - \sum_{k \to j} b_{k \to j} X_j)^2$. i.e. $\Theta_{ii} = 1 + \sum_j b_{i \to j}^2$.

Similarly, $\Theta_{ij}$ is the coefficient of $2X_i X_j$. We have a contribution of $b_{i \to j}$ if $i$ is a parent of $j$, coming from the sum of $(X_j - \sum_{k \to j} b_{k \to j} X_j)^2$, or $b_{j \to i}$ if $j$ is a parent of $i$, coming from the sum of $(X_i - \sum_{k \to i} b_{k \to i} X_j)^2$. Also, for each common child $l$ of $i, j$, we get a contribution $b_{i \to l} b_{j \to l}$ in the sum of $(X_l - \sum_{k \to j} b_{k \to l} X_l)^2$. Overall, $\Theta_{ij} = b_{i \to j} + b_{j \to i} + \sum_l b_{i \to l} b_{j \to l}$. $\square$

Now we combine these observations with a lemma in (Misra et al., 2020), which is stated for arbitrary undirected Gaussian Graphical Models. It quantifies the accuracy of estimating the coefficients in OLS for a node, if we regress it with a small subset of nodes that contains its Markov Blanket. We will apply it to the marginal distribution of $(X_i, X_T)$.

**Lemma A.5.** *(For general $L_0$ constrained sparse linear regression) (Proposition 2 in (Misra et al., 2020)) Suppose we have a multivariate $n$-dimensional zero-mean Gaussian model with inverse covariance matrix $\Theta$, and $m$ i.i.d. samples from this model, where*

$$m \geq 2d + \frac{8}{\varepsilon^2} d \log(n) + \frac{4}{\varepsilon^2} \log(\frac{2d}{\delta}),$$

*For a node $i$, let $\pi(i)$ be the neighbors of $i$ in the undirected graphical model induced by $\Theta$. Suppose $A \subseteq [n] \setminus \{i\}$ with $|A| \leq 2d$ and $\pi(i) \subseteq A$. Then the OLS coefficients $\hat{\beta}$ when we regress $X_i$ on $X_A$ using $m$ samples satisfy with probability at least $1 - \delta$ the following bound : $\forall j \in A$,*

$$\left| \hat{\beta}_{ij} - \frac{\Theta_{ij}}{\Theta_{ii}} \right| \leq \varepsilon \sqrt{1 + \frac{\Theta_{jj}}{\Theta_{ii}}}$$

We now make the connection with our DAG model more precise. As discussed earlier, let $i$ be some node and the set of the node before $i$ in the ordering is $T$. We consider the marginal distribution of $(X_T, X_i)$, which is still a Bayesnet. Let $\Theta$ be the inverse covariance matrix of $(X_T, X_i)$. Since $i$ is the last node in the ordering for this smaller Bayesnet, it is easy to see that its Markov Blanket $\pi(i)$, if the joint distribution is viewed as an undirected Graphical Model induced by $\Theta$, is exactly equal to the set of parents $\mathrm{pa}(i)$ in the DAG. This can be inferred for example from Lemma A.4 by noticing that $\Theta_{ij}$ is non-zero if and only if $b_{ji} \neq 0$ (since $i$ has no children in this marginal distribution).

This allows us to apply Lemma A.5 to find the parents of $i$ and gives rise to the following Corollary.

**Corollary A.6.** *Let $i$ be a node and $T$ be the subset of nodes that come before $i$ in the ordering. Suppose we use $m$ i.i.d. samples to regress $X_i$ on $X_J$, where $J \subseteq T$, $|J| \leq d$ and $\mathrm{pa}(i) \subseteq J$. Let $\hat{\beta}$ be the coefficients output by the regression. Suppose*

$$m \geq 64d \log(n) \cdot \tau \cdot b_{\min}^{-2}$$

*Then, with probability at least $1 - \delta$ we have that for all $j \in \mathrm{pa}(i)$,*

$$|\hat{\beta}_{ji}| > b_{\min}/2$$

*and for all $j \notin \mathrm{pa}(i)$*

$$|\hat{\beta}_{ji}| < b_{\min}/4$$

*Proof.* By the preceding observations, we can apply Lemma A.5 to analyze the regression. If $\Theta$ is the inverse covariance matrix of the joint distribution of $(X_i, X_T)$, then by Lemma A.4, since $X_i$ is the last node, we have $\Theta_{ij}$ is $b_{j \to i}$, $\Theta_{ii} = 1$ and $\Theta_{jj} = 1 + \sum_{j \prec k \preceq i} b_{j \to k}^2$.

Thus $|\Theta_{ij}/\Theta_{ii}| = |b_{ji}| \geq b_{\min}$ if $j \in \mathrm{pa}(i)$. Also, if $j \notin \mathrm{pa}(i)$, then $|\Theta_{ij}/\Theta_{ii}| = 0$. Thus, by applying Lemma A.5, we need to ensure $\varepsilon \sqrt{1 + \sum_{j \prec k \preceq i} b_{j \to k}^2} \leq b_{\min}/2$ for all $i, j$. So, it suffices to have $\varepsilon < b_{\min}/(2\sqrt{\tau})$. By union bound, we need, $\delta \leftarrow \delta/n$. To determine the sample complexity, we plug these values in Lemma A.5

$$m \geq 2d + \frac{8}{(b_{\min}/2\sqrt{\tau})^2} d \log n + \frac{4}{(b_{\min}/2\sqrt{\tau})^2} \log(2dn).$$

Where we take $m = 64d \log(n) \cdot \tau \cdot b_{\min}^{-2}$ suffices, and this finishes the proof. $\square$

Below we give the details of the proof of the second half of Theorem 1:

*Proof.* We first prove that for all $i$ and prior nodes $T$ in the topological ordering, Algorithm 1 correctly identifies a subset $J$ that contains $\mathrm{pa}(i)$. The strategy is similar to (Misra et al., 2020). For every candidate $J \subseteq T$ with $|T| = d$, we iterate over all other $K \subseteq T$ with $|K| = d$ and regress $X_i$ on $X_{J \cup K}$.

- Suppose $\mathrm{pa}(i) \subseteq J$. Then, for all other $K$, for each $j \in K$ we have $|\hat{\beta}_{ji}| < b_{\min}/2$ with prob $\geq 1 - \delta$ by Lemma A.6, so the condition in line 17 of Algorithm 1 doesn't hold and $J$ is accepted as a true superset of the parents.

- Suppose there is a $j \in \mathrm{pa}(i)$ with $j \notin J$. Then, there exists a $K$ with $j \in K$ and $\mathrm{pa}(i) \subseteq J \cup K$. Then, regressing with $J \cup K$ will yield $|\hat{\beta}_{ji}| > b_{\min}/2$ by Lemma A.6, leading to the condition in line 17 to be satisfied. Thus, set $J$ is rejected.

After we find a superset $J$ of the parents, we can simply remove the nodes with small $\beta$ coefficients in the regression and appeal to Lemma A.6 again for correctness in an identical fashion.

$\square$

# B. Proof of Lower Bound (Theorem 2.3)

The standard testing setting involves a family of distributions indexed by a set $\mathcal{V}$. A random variable $V$ is drawn uniformly at random from $\mathcal{V}$ and $X$ is generated according to the distribution corresponding to $V$. Note that $X$ could be one or multiple samples from this distribution. Then, an algorithm that takes as input $X$ and guesses $V$ is denoted by $\hat{V}$. We use Corollary 9.4.2 from (Duchi, 2016), which is a version of Fano's Inequality for testing.

**Lemma B.1.** *Assume that $V$ is uniform on $\mathcal{V}$. For any Markov chain $V \to X \to \hat{V}$,*

$$P(V = \hat{V}) \geq 1 - \frac{I(V; X) + \log 2}{\log |V|}.$$

In our case, the set $\mathcal{V}$ corresponds to a set of graphs $\mathcal{G} = \{G_1, G_2, \ldots, G_k\}$ that we would like to distinguish from each other. $V$ denotes a graph chosen uniformly at random among $\mathcal{V}$. Let $Q_V$ denote the distribution of one sample drawn from graph $V$ and $Q_V^{\otimes m}$ denotes the product distribution on $m$ i.i.d. samples drawn from $V$. The mapping $X \to \hat{V}$ is the estimator of the graph from the samples for a model. From Equation (9.4.5) in (Duchi, 2016), we have

$$I(V; X) \leq \frac{1}{|\mathcal{V}|^2} \sum_{V, V' \in \mathcal{V}} KL(Q_V^{\otimes m} \| Q_{V'}^{\otimes m}) = \frac{m}{|\mathcal{V}|^2} \sum_{G, G' \in \mathcal{G}} KL(Q_V \| Q_{V'})$$

(This is due to the tensorization of KL for product distributions.) Therefore, if we can construct a family of $|\mathcal{G}|$ graphs such that for any $G, G' \in \mathcal{G}$, we have $KL(Q_G \| Q_{G'}) \leq \alpha$, then we need at least $m \geq \frac{\log(|\mathcal{G}|)/2 - \log 2}{\alpha}$ samples to obtain prediction probability $\geq 1/2$. We consider the following ensembles:

**Ensemble for $\log(n)/b_{\min}^4$.** Assume 2 divides $n$, consider $\mathcal{G} = \{G_0, G_1, G_1, G_2, \ldots, G_{n/2}\}$. As Figure 4 shows, all graphs are defined on random variables $Y_1, Y_2, \ldots, Y_{n/2}$ and $Z_1, \ldots, Z_{n/2}$. $G_0$ is a direct matching of $Y_i \to Z_i$, with each edge having weight $b_{\min}$. $G_k$ is $k$th model and is equal to $G_0$, except that the edge between $Y_k$ and $Z_k$ is reversed.

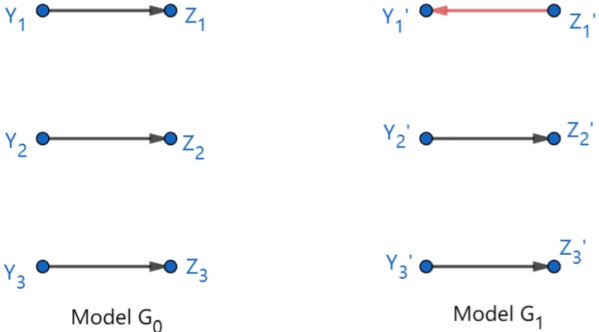

**Figure 4.** Schematic graph for the ensemble. Every $G_i$ for $j \geq 1$ is obtained from $G_0$ by reversing one edge in the matching of model $G_0$, e.g. $G_1$ is just reversing $Y_1 \to Z_1$ in $G_0$.

We notice that this model is a product measure of $n/2$ disjoint pairs of variables. Therefore, any two models differ in at most two edges. Let $(Y, Z)$ be the joint distribution of a two variable DAG with edge $Y \to Z$ with weight $b_{\min}$. Then, by the preceding observations, for any two models $G_i, G_j$, their KL divergence is no more then $2 \times KL((Y, Z) \| (Z, Y))$. Next, we calculate this distance.

First, the distribution of $(Y_1, Z_1)$ has the multivariate Gaussian distribution of covariance matrix

$$\Sigma_1 = \begin{pmatrix} 1 & b_{\min} \\ b_{\min} & 1 + b_{\min}^2 \end{pmatrix}.$$

Similarly, $(Z, Y)$ follows a multivariate Gaussian distribution of covariance matrix

$$\Sigma_2 = \begin{pmatrix} 1 + b_{\min}^2 & b_{\min} \\ b_{\min} & 1 \end{pmatrix}.$$

Therefore, the KL distance is

$$KL((Y,Z)||(Z,Y)) = \frac{1}{2}\left(\log\frac{|\Sigma_1|}{|\Sigma_2|} - 2 + \text{Tr}(\Sigma^{-1}\Sigma_2)\right) = \frac{1}{2}\left(-2 + \text{Tr}\begin{pmatrix} 1 + b_{\min}^2 + b_{\min}^4 & b_{\min}^3 \\ -b_{\min}^3 & 1 - b_{\min}^2 \end{pmatrix}\right) = \frac{1}{2}b_{\min}^4$$

So, in this case, any two models has KL distance at most $b_{\min}^4$, and this implies that with probability $\geq 1/2$ we need at least $(\log(n/2)/2 - \log 2)/b_{\min}^4 = \log(n/8)/(2b_{\min}^4)$ samples to find the right model.

**Ensemble for** $\log(n)\tau/b_{\min}^2$**.** Assume 3 divides $n$, consider $\mathcal{G} = \{G_0, G_1, G_1, G_2, \ldots, G_{n/3}\}$. Our graphs now consist of independent blocks on variables $X_i, Y_i, Z_i$. As Figure 5 shows, in $G_0$ the topology of each triplet is $X_i \to Y_i, X_i \to Z_i, Y_i \to Z_i$. The weights are: $X_i \to Z_i$ with $b_{\min}$ weight, $X_i \to Y_i$ with $B$ weight, and $Y_i \to Z_i$ with $b_1$ weight for some $B > b_1 > b_{\min} > 0$. For $G_i$ with $i > 1$, the topology is the same as $G_0$ but we change exactly one of the triangles $X_iY_iZ_i$: it becomes $X_i \to Z_i$ with no edge, $X_i \to Y_i$ with $B$ weight, and $Y_i$ to $Z_i$ with $b_1 + \frac{Bb_{\min}}{1+B^2}$ weight.

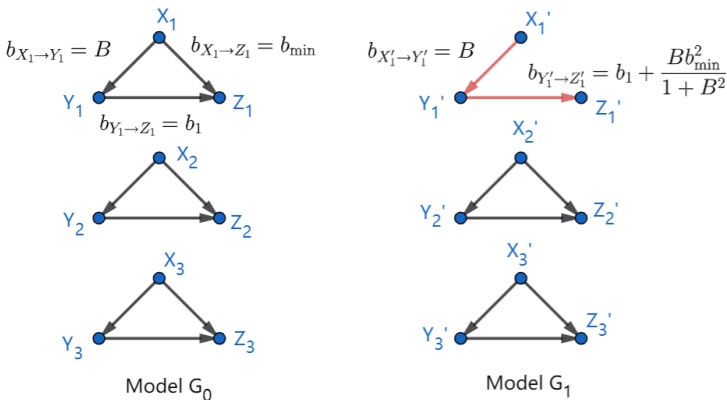

*Figure 5.* Schematic graph for the ensemble. Every model $G_i$ with $i \geq 1$ is obtained from $G_0$ by switching one triangle to a chain, i.e. $G_1$ is just changing the triangle $X_1Y_1Z_1$ to a chain $X_1 \to Y_1 \to Z_1$.

Let $(X, Y, Z)$ be the distribution of the DAG with topology $X \to Z$ with $b_{\min}$ weight, $X \to Y$ with $B$ weight, and $Y \to Z$ with $b_1$ weight. Let $(X', Y', Z')$ be the distribution of the DAG with topology $X' \to Y'$ with $B$ weight, and $Y' \to Z'$ with $b_1 + \frac{b_{\min}}{1+B^2}$ weight. Similar to the observations for the previous ensemble, the KL distance between any two models is bounded by $2 \cdot KL((X,Y,Z)||(X',Y',Z'))$.

The distribution of $(X, Y, Z)$ has the multivariate Gaussian distribution of covariance matrix

$$\Sigma_1 = \begin{pmatrix} 1 & B & Bb_1 + b_{\min} \\ B & 1+B^2 & b_1 + B^2b_1 + Bb_{\min} \\ Bb_1 + b_{\min} & b_1 + B^2b_1 + Bb_{\min} & 1 + (Bb_1 + b_{\min})^2 + b_1^2 \end{pmatrix}.$$

The distribution of $(X', Y', Z')$ is a multivariate Gaussian distribution of covariance matrix

$$\Sigma_2 = \begin{pmatrix} 1 & B & Bb_1 + \frac{B^2b_{\min}}{1+B^2} \\ B & 1+B^2 & b_1 + B^2b_1 + Bb_{\min} \\ Bb_1 + \frac{B^2b_{\min}}{1+B^2} & b_1 + B^2b_1 + Bb_{\min} & 1 + (b_1 + \frac{Bb_{\min}}{1+B^2})^2(B^2 + 1) \end{pmatrix}.$$

Therefore, we can calculate $|\Sigma_1| = |\Sigma_2| = 1$ (because it is the inverse of a product of two upper triangle matrix with diagonal 1: Notice that $|\Sigma_1| = |((I-B)^{-1})^\top (I-B)^{-1}| = |((I-B)^{-1})^\top| \cdot |(I-B)^{-1}| = 1 \cdot 1 = 1$) and

$$\Sigma_1^{-1}\Sigma_2 = \begin{pmatrix} 1 + \frac{b_{\min}^2}{1+B^2} & 0 & -b_{\min} \\ \frac{b_{\min}b_1}{1+B^2} & 1 & \frac{b_{\min}B}{1+B^2} \\ \frac{-b_{\min}}{1+B^2} & 0 & 1 \end{pmatrix}, \Sigma_2^{-1}\Sigma_1 = \begin{pmatrix} 1 & 0 & b_{\min} \\ -\frac{b_{\min}(b_{\min}B + b_1 + B^2b_1)}{(1+B^2)^2} & 1 & -\frac{b_{\min}(B + Bb_{\min}^2 + B^3 + b_{\min}b_1 + b_1b_{\min}B^2)}{(1+B^2)^2} \\ \frac{b_{\min}}{1+B^2} & 0 & 1 + \frac{b_{\min}^2}{1+B^2} \end{pmatrix}$$

Both of the traces are $3 + \frac{b^2_{\min}}{1+B^2}$. Therefore,

$$KL((X,Y,Z)||(X',Y',Z')) = \frac{1}{2}\left(\log\frac{|\Sigma_1|}{|\Sigma_2|} - 3 + \mathrm{Tr}(\Sigma^{-1}\Sigma_2)\right) = \frac{1}{2}\left(-3 + 3 + \frac{b^2_{\min}}{1+B^2}\right) = b^2_{\min}/(1+B^2)$$

So, in this case, any two models has KL distance at most $b^2_{\min}/(1+B^2) = b^2_{\min}/\tau$, and this implies that with probability $\geq 1/2$, we need at least $(\log(n/3)/2 - \log 2)\tau/b^2_{\min} = \log(n/12)\tau/(2b^2_{\min})$ samples to find the correct topology.

Thus, from both ensembles, Theorem 2.3 follows.

## C. Analysis of Efficient Algorithm (Theorem 2.6)

We give the full version for Algorithm 2 below.

---

**Algorithm 3** Efficient algorithm for recovering the topology

---

**input** Samples $(X^{(1)}, X^{(2)}, \ldots, X^{(m)})$
**output** Topology $\hat{G}$
1: $T \leftarrow \left[i : \widehat{\mathrm{Var}(X_i)} \leq (1 + b^2_{\min}/2)\right]$
2: **while** $|T| < n$ **do**
3:     $M = \emptyset$
4:     **for** $i \in [n]\backslash T$ **do**
5:         Choose $\lambda = O(\sqrt{\log(n/\delta)}R/\sqrt{m})$
6:         $\tilde{\beta} \leftarrow$ LASSO regression of $X_i$ on $X_T$
7:         $J \leftarrow \{k||\tilde{\beta}_k| \geq b_{\min}/2\}$, if $|J| > d$ then go to line 4.
8:         Perform OLS for $X_i$ on $X_J$, get coefficient $\hat{\beta}$.
9:         $MSE \leftarrow \frac{1}{m}\sum_j \left(X_i^{(j)} - X_J^{(j)}\hat{\beta}\right)^2$
10:        **if** $MSE \leq 1 + b^2_{\min}/2$ **then**
11:            $M \leftarrow M \cup \{i\}$
12:            $\mathrm{pa}(i) \leftarrow \{j \in J : |\hat{\beta}_j| \geq b_{\min}/2\}$
13:        **end if**
14:     **end for**
15:     $T \leftarrow T.\mathrm{append}(M)$
16: **end while**

---

In line 6, we perform the LASSO algorithm to find the next vertex for the topological order as well as to find the parents. For any node $i$ and subset $T$ that appears before $i$ in the ordering, we are optimizing the Lagrangian LASSO:

$$\hat{\beta} = \underset{\beta \in \mathbb{R}^{|T|}}{\mathrm{argmin}}\left\{\frac{1}{2N}\sum_{i=1}^{N}(X_i^{(j)} - \beta^\top X_T^{(j)})^2 + \lambda\|\beta\|_1\right\} \tag{10}$$

For a vector $\beta$ and a subset $S$ of coordinates, we use $\beta_S$ to denote the subvector corresponding to the coordinates in $S$. As usual, let $\beta^*$ be the population coefficients of $X_i$ with respect to $X_T$. The following Theorem from (Wainwright, 2019) provides guarantees about the estimation error of the LASSO.

**Lemma C.1.** *(Combination of Theorems 7.18, 7.19 in (Wainwright, 2019)) Consider the Lagrangian Lasso where $X_T$ is generated i.i.d. from $\mathcal{N}(0, \Sigma)$. The independent noise for each sample is $w^{(j)} = X_i^{(j)} - \beta^{*\top}X_T^{(j)}$. Choose the regularization parameter $\lambda \geq \frac{2}{N}\max_{k \in T}\left|\sum_{j=1}^{m} w^{(j)}X_k^{(j)}\right|$. For any $\theta^* \in \mathbb{R}^d$, with probability at least $1 - \frac{e^{-m/32}}{1-e^{-m/32}}$ any optimal solution $\hat{\theta}$ satisfies the bound*

$$\left\|\hat{\beta} - \beta^*\right\|_2^2 \leq 9216\frac{\lambda^2}{\bar{\kappa}^2}|S| + 128\frac{\lambda}{\bar{\kappa}}\|\beta^*_{S^c}\|_1 + 12800\frac{\rho^2(\Sigma)}{\bar{\kappa}}\frac{\log n}{m}\|\beta^*_{S^c}\|_1^2 \tag{11}$$

*valid for any subset $S$ with cardinality $|S| \leq \frac{1}{25600} \frac{\bar{\kappa}}{\rho^2(\Sigma)} \frac{m}{\log n}$. Here, $\rho^2(\Sigma)$ is the maximum diagonal entry of $\Sigma$ and $\bar{\kappa}$ is the smallest eigenvalue of $\Sigma$.*

There is a twist in the algorithm after computing the LASSO. First of all, if the output vector has more than $d$ "large" coefficients, this clearly means that $T$ does not contain the full neighborhood and thus $i$ is not the next node in the ordering. If it has at most $d$ "large" coefficients, instead of just calculating the MSE using the output of the LASSO, we run an additional OLS to verify whether $i$ is the next node in the ordering. Then, we will use the property A.1 to judge whether we have the correct parents. Specifically, if $i$ is the next node after $T$, then LASSO will find the correct parent set $J$ (**not** a superset) and the expected MSE after the OLS is $1$. If $i$ is not the next node after $T$, then for any $J \subseteq T$, the expected MSE after the OLS is $1 + b_{\min}^2$. Since the sample complexity for Theorem 2.1 is upper bounded by that of 2.6, this verifying process is going to be successful with high probability: we can see this by repeating the proof of Theorem 2.1. The reason we use OLS instead of Lasso is that it is harder to argue about the MSE of Lasso when not all parents are involved.

Therefore, the only task is to guarantee that LASSO will determine the true parents for any $i$ that is next in the topological order of $T$. In order to have that, we need the $\mathcal{L}_2$ norm of the distance of $\tilde{\beta}$ and $\beta^*$ to be smaller than $b_{\min}/2$. Applying Lemma C.1 above for LASSO in our algorithm, we need to select $\lambda$ and $m$ appropriately for this to hold. Therefore, it suffices to choose $\lambda, m$ so that the following inequalities hold simultaneously:

$$\frac{1}{25600} \frac{\bar{\kappa}}{\rho^2(\Sigma)} \frac{m}{\log n} \geq |S| \geq d \tag{12}$$

$$9216 \frac{\lambda^2}{\bar{\kappa}^2} |S| \leq \frac{b_{\min}^2}{16} \tag{13}$$

$$128 \frac{\lambda}{\bar{\kappa}} W_1 \leq \frac{b_{\min}^2}{16} \tag{14}$$

$$12800 \frac{R^2 \log n}{\bar{\kappa} N} W_1^2 \leq \frac{b_{\min}^2}{16} \tag{15}$$

Equation (12) requires $|S|$ to be large enough, and Equations (13), (14) and (15) require $\left\| \tilde{\beta} - \beta^* \right\|^2$ smaller than $b_{\min}^2/4$, which achieves the guarantee. Also, here we introduce $W_1$ to be the maximum $\mathcal{L}_1$ norm of the coefficient vector corresponding to the parents of each node: that is,

$$W_1 = \max_i \sum_{j \to i} |b_{j \to i}|.$$

Now, we bound the variables one by one. By our assumption, we have $\rho^2(\Sigma) < R^2$, $W_1 \leq d\sqrt{\tau}$ (because $b_{i \to j} \leq \sqrt{\tau}$) and by Cauchy-Schwartz inequality, $\bar{\kappa} \geq \frac{1}{(d+1)\cdot\tau} \geq \frac{1}{2d\cdot\tau}$. We apply the following lemma for all $X_T$ since the sub-Bayesnet has the same model as the whole Bayesnet:

**Lemma C.2.** *Assume we have the Bayesnet as Equation (17). Let $\bar{\kappa}$ is the smallest eigenvalue of the covariance matrix of $X_1, X_2, \ldots, X_n$. Then, we have $\bar{\kappa} \geq \frac{1}{(d+1)\tau}$.*

*Proof.* We argue by Cauchy-Schwartz inequality. Let $\Sigma$ be the covariance matrix of $X_1, X_2, \ldots, X_n$, and $\Theta = \Sigma^{-1}$. So the least eigenvalue of $\Sigma$ is the largest eigenvalue of $\Theta$. Let $x = (x_1, x_2, \ldots, x_n)$ and $||x|| = 1$, so that we have $x^T \Theta x = \sum_{i=1}^n (\sum_{j=1}^n x_i - b_{j \to i} x_j)^2$ (by the proof of Lemma A.4.) By Cauchy-Schwartz inequality, we have $\sum_{i=1}^n (\sum_{j=1}^n x_i - b_{j \to i} x_j)^2 \leq \sum_{i=1}^n (d+1) \sum_{j=1}^n x_j^2 (1 + \sum_{i=1}^n b_{i \to j}^2) \leq (d+1)\tau$. Thus, we have $\|\Theta\|_{op} \leq (d+1)\tau$. Since $\Sigma = \Theta^{-1}$, we have the least eigenvalue of $\Sigma$ is no less than $\frac{1}{(d+1)\tau}$. $\qquad\square$

Also, to bound $\lambda$, we need the following Lemma, whose proof is given in Section D.5.

**Lemma C.3.** *Let $Z_i$ be a random variable of $X_i \cdot Y_i$ where $X_i, Y_i$ are independent $\mathcal{N}(0, 1)$. Then, there is a universal constant $C_\lambda = 32$, such that for any $\varepsilon < 1/2, \delta < 1/2$ given $m \geq C_\lambda(\log(2/\delta))/\varepsilon^2$, we have that the following holds:*

$$\mathbb{P}\left(\left|\frac{1}{m}\sum_{i=1}^m Z_i\right| > \varepsilon\right) \leq \delta$$

First, we calculate an $m$ to make all inequalities hold for $|S| \geq d$ that satisfy equation (12). This requires that $m \geq 51200 \cdot d^2 \cdot \tau \cdot R^2 \cdot \log(n)$.

Also, since we have assumed that for each $i$, the variance of $X_i$ is $\leq R$, thus, for each $k$, $\sum_{j=1}^m w^{(j)} X_k^{(j)}$ has each $w^{(j)}$ is sampled independently from $\mathcal{N}(0,1)$ and $X_k^{(j)}$ is sampled independently from $\mathcal{N}(0, \mathrm{Var}(X_k))$, where $\mathrm{Var}(X_k) \leq R^2$. Therefore, because we need to execute $n^2$ LASSOs, so we need to substitute $\delta \leftarrow \delta/n^2$ in Lemma C.3, in order to apply a union bound. Therefore, by Lemma C.3 and union bound, we substitute $\varepsilon \leftarrow \lambda/(2R)$ and $\delta \leftarrow \delta/4n^2$ (the reason for the extra factor of $4$ is to accommodate some failure probability for the accuracy of $r, VE, BE$ and for the failure of Lemma C.1). Thus, it suffices to have $m \geq 128 \log(8n^2/\delta)/\lambda^2$. Equivalently, this can be written as $\lambda \leq \frac{\sqrt{128 \log(8n^2/\delta)} R}{\sqrt{m}}$

Solving (13), (14), (15) and substituting the bound for $\bar\kappa, |S|, W_1$, we have

$$9216 \frac{128 \log(8n^2/\delta) R^2}{m} (4d\tau)^2 d \leq \frac{b_{\min}^2}{16}$$

$$128 \frac{\sqrt{128 \log(8n^2/\delta)} R}{\sqrt{m}} d^2 \tau^{1.5} \leq \frac{b_{\min}^2}{16}$$

$$12800 \frac{R^2 \log n}{m} d^3 \tau^2 \leq \frac{b_{\min}^2}{16}$$

Thus, we can choose $m \geq 2^{32} \cdot \log(n/\delta) \cdot d^4 \cdot \tau^3 \cdot R^2 \cdot b_{\min}^{-4}$ to satisfy the bounds above. This finishes the proof for Theorem 2.6.

## D. Additional Proofs

### D.1. Proof for Lemma 2.2

By Lemma A.4, the diagonal entry of $\Theta$ is

$$\Theta_{ii} = 1 + \sum_{i \to j} b_{i \to j}^2$$

So, by choosing $x$ to be a standard basis vector, we get that there exist $x_1, x_2$ such that

$$x_1^\top \Theta x_1 = \min_i |\Theta_{ii}| \quad , \quad x_2^\top \Theta x_2 = \max_i |\Theta_{ii}|$$

It is clear that

$$\max_i |\Theta_{ii}| \geq \tau$$

by definition. Also, by choosing the diagonal element of the last node in the topological ordering, it is clear that

$$\min_i |\Theta_{ii}| = 1$$

Therefore, the condition number of $\Theta$ is at least $\tau$, and the same holds for $\Sigma$.

### D.2. Proof for Lemma 2.5

**For (1)** Let $\Sigma$ be the covariance matrix, then consider all unit vector $x$, $x^\top \Sigma x$ ranges from 1 (choose the first vertex in the topological order) to $\max_i \mathrm{Var}(X_i) = R$. Also, we know that the smallest and largest eigenvalues of $\Sigma$ ($\lambda_{\min}$ and $\lambda_{\max}$) satisfy that $\lambda_{\min} = \min_{\|x\|=1} x^\top \Sigma x$ and $\lambda_{\max} = \max_{\|x\|=1} x^\top \Sigma x$. So we have $\lambda_{\min} \leq 1$ and $\lambda_{\max} \geq R$ and thus the condition number of $\Sigma$ is at least $R$.

**For (2)** We consider a directed rooted tree with branching factor 2 with variables $(X_1, \ldots, X_n)$. Consider root $X_1 \sim \mathcal{N}(0,1)$. And for any $i$, $X_i$ has two children $X_{2i+1}$ and $X_{2i+1}$ with recurrence $X_{2i} = \lambda x_i + \varepsilon_{2i}$ and $X_{2i+1} = \lambda x_i + \varepsilon_{2i+1}$ where $\varepsilon_{2i}, \varepsilon_{2i+1}$ are i.i.d. $\mathcal{N}(0,1)$. Assume the tree has height $h$, and thus $n = 2^{h+1} - 1$. Choose some $\lambda \in (\frac{1}{\sqrt{2}}, 1)$. For any leaf ($X_i$ for $i \geq 2^h$), we can write $X_i = \varepsilon_i + \lambda \varepsilon_{\lfloor i/2 \rfloor} + \lambda^2 \varepsilon_{\lfloor i/4 \rfloor} + \cdots + \lambda^h \varepsilon_{\lfloor i/2^h \rfloor}$. Therefore, its variance is $1 + \lambda^2 + \cdots + \lambda^{2h} < \frac{1}{1-\lambda^2}$.

On the other hand, to lower bound the condition number, we need to find two different values of $x^\top \Sigma x$ for some $\|x\| = 1$. If we choose $x$ to be the unit vector with $X_1$ coordinate to be $1$, the variance is $1$. We can also choose $x$ to be $1/\sqrt{2^h}$ for coordinates corresponding to leaves and $0$ otherwise. Then, $x^\top \Sigma x$ corresponds to the variance of the linear combination $\frac{1}{\sqrt{2^h}}(X_{2^h} + \cdots + X_{2^{h+1}-1})$. We are going to prove that the variance is

$$\text{Var}(X_{2^h} + \cdots + X_{2^{h+1}-1}) = 1 + (2\lambda^2) + (2\lambda^2)^2 + \cdots + (2\lambda^2)^h. \tag{16}$$

If this holds, the condition number is at least $\frac{(2\lambda^2)^{h+1}-1}{(2\lambda^2-1)(1-\lambda)} \geq \frac{n^{\log_2(2\lambda^2)}-1}{(2\lambda^2-1)(1-\lambda)}$. By choosing $\lambda = 2^{(\alpha-1)/2}$ we can get the desired bound.

Now we prove (16). Suppose $V_\ell$ is the conditional variance of $\frac{1}{\sqrt{2^\ell}}(X_{2^\ell} + \cdots + X_{2^{\ell+1}-1})$ conditioned on $X_1$, for any value of $\ell$. We have $V_1$ to be the conditional variance of $\frac{1}{\sqrt{2}}(X_2 + X_3)$ on $X_1$, which is $2/2 = 1$. We consider a recurrence. By the law of total variance

$$\text{Var}(\frac{1}{\sqrt{2}^h}(X_{2^h} + \cdots + X_{2^{h+1}-1})|X_1)$$
$$= \text{Var}(\frac{1}{\sqrt{2}^h}(X_{2^h} + \cdots + X_{2^{h+1}-1})|X_2, X_3) + \text{Var}(\frac{1}{\sqrt{2}^h}(\mathbb{E}[X_{2^h} + \cdots + X_{2^{h+1}-1}|X_2, X_3]|X_1)$$

The first part we can write as

$$\frac{1}{2}\text{Var}(\frac{1}{\sqrt{2}^{h-1}}(X_{2^h} + \cdots + X_{2^h+2^{h-1}-1} + X_{2^h+2^{h-1}} + \ldots, X_{2^{h+1}-1})|X_2, X_3)$$

Notice that $X_{2^h}, \ldots, X_{2^h+2^{h-1}-1}$ are the leaves of the subtree rooted at $X_2$, and $X_{2^h+2^{h-1}}, \ldots, X_{2^{h+1}-1}$ are the leaves of the subtree rooted at $X_3$. Both subtrees with root $X_2$ and with root $X_3$ have the same topology as a tree of height $h-1$. So this term can be written as

$$\frac{1}{2}\text{Var}(\frac{1}{\sqrt{2}^{h-1}}(X_{2^h} + \cdots + X_{2^h+2^{h-1}-1} + X_{2^h+2^{h-1}} + \ldots, X_{2^{h+1}-1})|X_2, X_3)$$
$$= \frac{1}{2}(\text{Var}(\frac{1}{\sqrt{2}^{h-1}}(X_{2^h} + \cdots + X_{2^h+2^{h-1}-1}|X_2)) + \text{Var}(\frac{1}{\sqrt{2}^{h-1}}(X_{2^h+2^{h-1}} + \ldots, X_{2^{h+1}-1})|X_3))$$
$$= \frac{1}{2}(V_{h-1} + V_{h-1}) = V_{h-1}$$

For the second term, we have for every $2^h \leq i < 2^h + 2^{h-1}$, $\mathbb{E}[X_i|X_2] = \lambda^{h-1}X_2$ (the other noise on this branch is mean zero, and the coefficient of $X_2$ is $\lambda^{h-1}$.) For identical reasons, for every $2^h + 2^{h-1} \leq i < 2^{h+1}$, $\mathbb{E}[X_i|X_3] = \lambda^{h-1}X_3$. Therefore, we have

$$\text{Var}(\frac{1}{\sqrt{2}^h}(\mathbb{E}[X_{2^h} + \cdots + X_{2^{h+1}-1}|X_2, X_3]|X_1)$$
$$= \text{Var}(\frac{1}{\sqrt{2}^h}(\mathbb{E}[X_{2^h} + \cdots + X_{2^h+2^{h-1}-1}|X_2] + \mathbb{E}[X_{2^h+2^{h-1}} + \ldots, X_{2^{h+1}-1}|X_3])|X_1)$$
$$= \text{Var}(\frac{1}{\sqrt{2}^h}(2^h\lambda^{h-1}X_2 + 2^{h-1}\lambda^{h-1}X_3)|X_1) = (2\lambda^2)^{h-1} \times \frac{1}{2}\text{Var}(V_2 + V_3|V_1) = (2\lambda^2)^{h-1}$$

So, by induction, we have $V_h = 1 + (2\lambda^2)^2 + \cdots + (2\lambda^2)^{h-1}$. Finally, for all $2^h \leq i < 2^{h+1}$, we have $\mathbb{E}[X_i|X_1] = \lambda^h X_1$.

This yields

$$\text{Var}(\frac{1}{\sqrt{2^h}}(X_{2^h} + \cdots + X_{2^{h+1}-1}))$$

$$= V_h + \text{Var}(\frac{1}{\sqrt{2^h}}\mathbb{E}[(X_{2^h} + \cdots + X_{2^{h+1}-1})]|X_1)$$

$$= V_h + \text{Var}(\frac{1}{\sqrt{2^h}} \times (2\lambda)^h X_1) = V_h + (2\lambda^2)^h = 1 + (2\lambda^2) + \cdots + (2\lambda^2)^h \ .$$

This finishes the proof.

### D.3. Proof of Lemma A.2

We will use the following standard result about concentration of the chi-square distribution.

**Lemma D.1.** *For Chi-square distribution, we have the following concentration inequality: if $t \sim \chi_k^2$*

$$\mathbb{P}(t > (1+\varepsilon)k) \leq (\frac{1+\varepsilon}{e^\varepsilon})^{k/2}$$

*and*

$$\mathbb{P}(t < (1-\varepsilon)k) \leq ((1-\varepsilon)e^\varepsilon)^{k/2}$$

This is a folklore bound and can be proved by using the MGF of the chi-squared distribution. Now we present the proof of A.2.

*Proof of Lemma A.2.* We know that $\hat{\beta} - \beta^*$ is a multivariate zero-mean Gaussian with covariance matrix $(\sum_{j=1}^m X_J^{(j)} X_J^{(j)\top})^{-1}$. This follows from the closed form solution of OLS. So we know that the sum

$$\sum_{j=1}^m \left( \left(X_J^{(j)}\right)^\top (\beta_J^* - \hat{\beta}_J) \right)^2 = (\beta_J^* - \hat{\beta}_J)^\top \sum_{j=1}^m X_J^{(j)} X_J^{(j)\top} (\beta_J^* - \hat{\beta}_J)$$

follows a $V^2\chi_k^2$ distribution. By Lemma D.1, take $\varepsilon \leftarrow 4\frac{\log(1/\delta)}{k} + 1$, we have

$$\mathbb{P}(t > (1+\varepsilon)k \leq (\frac{1+\varepsilon}{e^\varepsilon}))^{k/2} = \left( \frac{2 + 4\log(1/\delta)/k}{(1/\delta^{4/k})e} \right)^{k/2} = \delta \cdot (\frac{2 + 4\log(1/\delta)/k}{e/\delta^{2/k}})^{k/2}$$

Let $x = (1/\delta)^{2/k}$, we just need that $2 + 2\ln(x) \leq ex$. This inequality holds for any $x > 1$ and can be proved by taking derivatives. So accordingly, $m$ can be $(4\log(1/\delta) + 2k)/\varepsilon$. $\square$

### D.4. Proof for Lemma A.3

We will use Lemma D.1. We know that $\sum_{i=1}^m Y_m^2$ follows the distribution $\sigma^2\chi_m^2$, so we just choose $m$ such that the probability of $\chi_m^2 \geq (1+\varepsilon)m$ and $\chi_m^2 \geq (1+\varepsilon)m$ are both at most $\delta/2$. Since $(1-\varepsilon)e^\varepsilon < \frac{1+\varepsilon}{e^\varepsilon}$, we can focus on the upper bound. We need $m$ to be $(\frac{1+\varepsilon}{e^\varepsilon})^{m/2} < \delta/2$, which gives $m \geq \frac{2\log(2/\delta)}{\varepsilon - \ln(1+\varepsilon)}$. Since $\varepsilon < 1/2$, we have $\varepsilon - \ln(1+\varepsilon) > \frac{1}{3}\varepsilon^2$. So we have $m \geq 6\log(2/\delta)/\varepsilon^2$.

### D.5. Proof for Lemma C.3

We cite a lemma of (van de Geer & Lederer, 2013):

**Lemma D.2.** *(Theorem 1) Let $X_1, \ldots, X_n$ be independent random variables with values in $\mathbb{R}$ and with mean zero. Suppose that for some constants $\sigma$ and $K$, one has*

$$\frac{1}{n}\sum_{i=1}^n \mathbb{E}|X_i|^m \leq \frac{m!}{2}K^{m-2}\sigma^2, \quad m = 2, 3, \ldots$$

*Then for all $t > 0$,*

$$\mathbb{P}\left(\frac{1}{\sqrt{n}}\left|\sum_{i=1}^{n} X_i\right| \geq \sigma\sqrt{2t} + \frac{Kt}{\sqrt{n}}\right) \leq 2\exp(-t).$$

First we calculate the moments of $Z_i$. We can easily calculate $\mathbb{E}|Z_i|^k = (\mathbb{E}|X_i|^k)^2 = \frac{2^k \Gamma(\frac{k+1}{2})^2}{\pi}$. Let $F(k) = \frac{2^k \Gamma(\frac{k+1}{2})^2}{\pi \cdot k!}$. We have $F(2) = \frac{1}{2}$, $F(3) = \frac{4}{3\pi} < 1/2$, and we also have that $F(k+2)/F(k) = \frac{k}{k+2} < 1$, so we have $F(k) < 1/2$ for all integers $k \geq 2$. This establish $\sigma = 1, K = 1$ in the Lemma. Therefore, by the lemma we have

$$\mathbb{P}\left(\frac{1}{m}\left|\sum_{i=1}^{m} Z_i\right| \geq \sqrt{\frac{2t}{m}} + \frac{t}{m}\right) \leq 2\exp(-t).$$

We take $t = \log(2/\delta)$ and $m$ such that $\sqrt{\frac{2t}{m}} + \frac{t}{m} \leq \varepsilon$. Since $\varepsilon < 1/2$, it suffice that $\sqrt{\frac{2t}{m}} < \varepsilon/4$, which implies $m \geq 32\log(2/\delta)/\varepsilon^2$ suffices.

### D.6. Comparison with (Gao et al., 2022)

As mentioned in Section 2.4, our lower bound is at least as good as the one provided in (Gao et al., 2022). To argue about that, we need to prove that $M^2 - 1 \geq db_{\min}^2$, where $\kappa \leq \sqrt{M}$.

Without loss of generality, we assume the variance to be 1, or otherwise, we just scale the covariance matrix Recall that, the adjacency matrix $B = \{b_{ij}\}$ is lower triangular after sorted by topological order, and the inverse covariance matrix is $\Theta = (I - B)(I - B)^\top$. Therefore, the inverse covariance matrix is having 1 as its determinant (because the determinant of both $I_B$ and $(I - B)^\top$ are 1 since they are lower/upper triangular), and thus the smallest eigenvalue is at most 1.

On the other hand, suppose $X_i$ has $d$ parents $X_{j_1}, \ldots, X_{j_d}$. We can write $X_i = \sum_{k=1}^{d} X_{j_k} b_{j_k \to i} + \varepsilon$. We take $X_i^2 + \sum_{k=1}^{d} X_{j_k}^2 = 1$. We can lower bound that

$$X^\top \Theta X = \sum_{i=1}^{n}\left(X_i - \sum_{j \to i} b_{j \to i} X_j\right)^2 \geq \left(X_i - \sum_{k=1}^{d} X_{j_k} b_{j_k \to i}\right)^2.$$

By Cauchy Schwartz, we have the upper bound

$$\left(X_i - \sum_{k=1}^{d} X_{j_k} b_{j_k \to i}\right)^2 \leq \left(1 + \sum_{k=1}^{d} b_{j_k \to i}^2\right)\left(X_i^2 + \sum_{k=1}^{d} X_{j_k}^2\right) = 1 + \sum_{k=1}^{d} b_{j_k \to i}^2.$$

This inequality can be achieved if

$$X_i = \frac{1}{\sqrt{1 + \sum_{k=1}^{d} b_{j_k \to i}^2}}, \quad X_{j_k} = \frac{-b_{j_k \to i}}{\sqrt{1 + \sum_{k=1}^{d} b_{j_k \to i}^2}} \quad \forall 1 \leq k \leq d.$$

So, we proved that the maximum eigenvalue of $\Theta$ is at least $1 + \sum_{k=1}^{d} b_{j_k \to i}^2 \geq 1 + db_{\min}^2$. Therefore, since the least eigenvalue of $\Theta$ is at most 1, the ratio between the maximum and minimum eigenvalue of $\Theta$ is at least $1 + db_{\min}^2$, or, the condition number is at least $1 + db_{\min}^2$. So, we have concluded $M \geq \sqrt{1 + db_{\min}^2}$.

## E. Discussion and Extension.

### E.1. Identifiability for non-equal variance

By calculating the conditional variance, one can identify the topological order of the DAG and thus identify the topology. However, we here give an example where the equal variance assumption is violated and one cannot find the topology of the DAG, even up to the same Markov Equivalence Class.

Let $b$ be a small positive number. Consider two different DAGs $G_1, G_2$, on variables $X, Y, Z$ and $X', Y', Z'$, respectively. The topology of $G_1$ is $X \to Y, Y \to Z$ and the topology of $G_2$ is $X' \to Z', Z' \to Y', X' \to Y'$. These two DAGs belong in different Markov Equivalence Classes. The variables in $G_1$ satisfy the relations with equal variance:

$$X = \varepsilon_1, Y = bX + \varepsilon_2, Z = bY + \varepsilon_3$$

where $\varepsilon_1, \varepsilon_2, \varepsilon_3$ are drawn i.i.d. from $\mathcal{N}(0, 1)$ distribution. Therefore, we can calculate the covariance matrix of the joint distribution $(X, Y, Z)$ as

$$\text{Cov}(X, Y, Z) = \begin{pmatrix} 1 & b & b^2 \\ b & b^2 + 1 & b^3 + b \\ b^2 & b^3 + b & b^4 + b^2 + 1 \end{pmatrix}$$

On the other hand, the variables in $G_2$ satisfy

$$X = \varepsilon_1', Z' = b^2 X + \varepsilon_2, Y' = \frac{b}{b^2 + 1} X' + \frac{b}{b^2 + 1} Z' + \varepsilon_3,$$

where $\varepsilon_1, \varepsilon_2, \varepsilon_3$ are i.i.d. centered Gaussians, but the variances are $1, b^2 + 1, \frac{1}{b^2 + 1}$ respectively. We can calso check that the covariance matrix of the joint distribution $(X', Y', Z')$ is

$$\text{Cov}(X', Y', Z') = \begin{pmatrix} 1 & b & b^2 \\ b & b^2 + 1 & b^3 + b \\ b^2 & b^3 + b & b^4 + b^2 + 1 \end{pmatrix}$$

It follows that $(X, Y, Z)$ and $(X', Y', Z')$ follow the same distribution, hence the two DAGs are indistinguishable. Notice that the minimum strength of the edges among the two networks is $b^2$, whereas the ratio of the variances in the second network is $1 + O(b^2)$. Thus, we have concluded our example to show a pair of graph that is not distinguishable if the variance ratio is at least $1 + O(b_{\min})$.

### E.2. Comparison to the PC Algorithm

A standard approach for structure recovery is the PC algorithm (Spirtes et al., 2001), which is based on conditional independence testing. In order to provide rigorous guarantees for correct recovery, most works rely on some form of the strong faithfulness assumption. It has been noted in prior work (Uhler et al., 2013) that this assumption might be too restrictive. Indeed, we next demonstrate that this assumption need not hold in our setting.

More formally, the strong faithfulness assumption requires that for any nodes $a, b$ and subset $S \subseteq V$, if, by the graph's structure, $a, b$ are not independent conditioning on $S$, then the conditional correlation of $a, b$ conditioning on $S$ is lower bounded. However, in our model this does not necessarily hold. We give the following example: consider the graph with three nodes $X, Y, Z$, and we have the model as

$$X = \varepsilon_1, Y = bX_1 + \varepsilon_2, Z = bY - b^2 X + \varepsilon_3$$

Here, $\varepsilon_1, \varepsilon_2, \varepsilon_3$ are i.i.d. $\mathcal{N}(0, 1)$. We can derive that $Z$ and $X$ are independent, as we could write $Z = b\varepsilon_2 + \varepsilon_3$. This happens due to path cancellation.

We complement the theoretical observations with experiments that show how our method compares to the PC algorithm. We use the PC algorithm package in `causal-learn` (Zheng et al., 2024). We use the same setting as in the simulation experiments of Section 4, with the hyperparameters $n = 5, 10, d = 2, 3$, and with $b_{\max} = 1$. We set 100 trials and the figure shows how much portion of graphs are correctly identified. All the graphs are random graphs. We note that in general, the PC algorithm is only guaranteed to find the correct Markov Equivalence Class, since some orientations depend on the order of increasing variance, which is extra information about the model that PC doesn't use. Therefore, we consider that the PC algorithm correctly identifies the graph if it outputs any graph in the same Markov Equivalence Class. This means that it detects all colliders (i.e. a child with at least two parents) and the skeleton (edges without directions). In Figure 6, we present the percentage of times that the PC correctly identified the Markov Equivalence Class, as well as the percentage of times that it identifies the correct skeleton, which is a relaxed goal. The result is as follows.

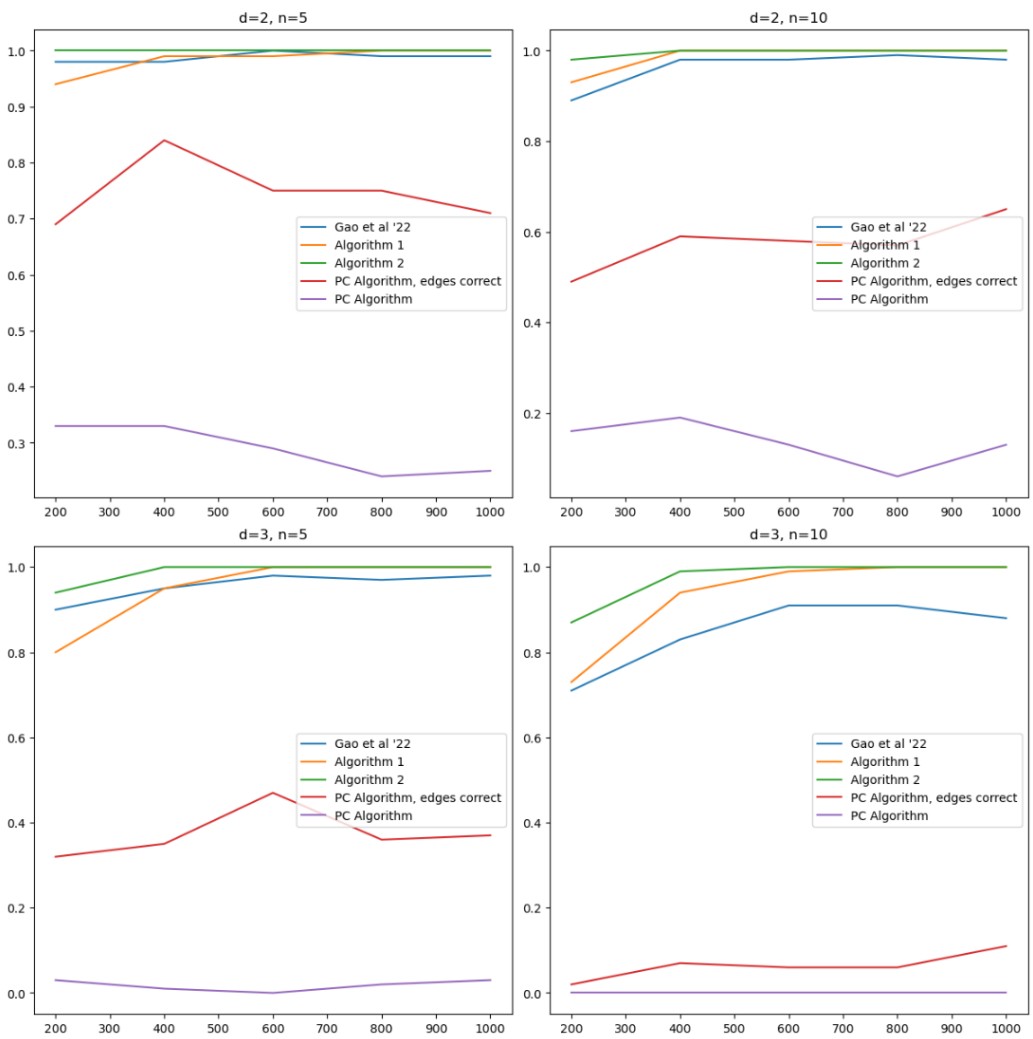

*Figure 6.* Comparison with the PC algorithm

In Figure 6, we notice that the PC algorithm behaves poorly even for the task of finding the skeleton of the graph, if the number of nodes becomes larger. This is explained by the observaton that the faithfulness assumption is less likely to hold if the number of nodes is larger. Note that we generate the edge weights with random signs, so as the number of nodes increases, the possibility of cancellations, such as the ones we saw in our previous example, also increases.

### E.3. Adaptivity of Algorithm 1: the case of unknown $\sigma^2, \beta_{\min}$ or $d$

One reasonable question concerns the situation when one or more parameters of the problem is unknown when we run Algorithm 1. Below, we explain some simple modifications that ensure the algorithm adapts to the unknown parameters. Essentially, we would like to argue that even if we do not the parameters a priori, if the algorithm is given enough samples it will succeed in identifying the graph.

**Unknown $\sigma^2$.** We notice that with the equal-variance assumption, $\min_i \mathrm{Var}(X_i) = \sigma^2$. Therefore, we can find the variable with empirical smallest variance, and take the empirical variance as $\hat{\sigma}^2$. As we're given $O(b_{\min}^{-4})$ samples, we can have with high probability, $\hat{\sigma}^2/\sigma^2 - 1$ to be $\pm b_{\min}^2/10$ (proven by Lemma A.3.) Thus, the error is small enough to make the Proposition A.1 hold, by slightly shrinking the threshold of judging whether a node is the immediate parent. In particular, the following modification of Proposition A.1 holds, with a similar proof.

**Proposition E.1** (with unknown $\sigma^2$). *(1) If $T$ contains all the parents, then there exists some $J \subseteq T$ and $|J| \leq d$ such that*

the conditional variance of $X_i|x_J$ is $\sigma^2$. With high probability, it is no more than $\hat{\sigma}^2(1 + b_{\min}^2/10)$.

*(2) If $T$ does not contain all the parents, then there exists a parent that is not in $T$, so for any coefficient vector $\beta_T$ each $X_i - X_T\beta_T$ is a Gaussian random variable that has variance at least $(1 + b_{\min}^2)\sigma^2$. Specifically, for any $J \subseteq T$, for any coefficient $\beta_J$, $X_i - X_J\beta_J$ also has variance $\geq 1 + b_{\min}^2$. With high probability, it is no less than $\hat{\sigma}^2(1 + 9b_{\min}^2/10)$.*

Therefore, we can assume that $\sigma^2$ is known and furthermore assume $\sigma^2 = 1$.

---

**Algorithm 4** Estimating the DAG Information Theoretically

---

**input** Samples $(X^{(1)}, X^{(2)}, \ldots, X^{(m)})$
**output** Topology $\hat{G}$
1: **Phase 1: Finding the Topological Ordering, not knowing $d$ and $b_{\min}$.**
2: $T = []$
3: Find $X_1 = \operatorname{argmin}_X \widehat{\operatorname{Var}}(X)$ and put $i \in T$
4: $\hat{d} = 1$
5: **while** $|T| < n$ **do**
6:    $M \leftarrow \emptyset, \text{FOUND} \leftarrow False$
7:    **for** $i \in [n]\backslash T$ **do**
8:      **for** $J \subseteq T, |J| \leq \hat{d}$ **do**
9:        $\beta_J \leftarrow$ Regress $X_i$ on $X_J$
10:        $MSE \leftarrow \frac{1}{m}\sum_j \left(X_i^{(j)} - X_J^{(j)}\beta_J\right)^2$
11:        **if** $MSE \leq 1 + O(\sqrt{\frac{\hat{d}\log n}{m}})$ **then**
12:          $T \leftarrow T.\text{append}(i)$
13:          $\text{FOUND} \leftarrow True$
14:        **end if**
15:      **end for**
16:    **end for**
17:    **if** $\text{FOUND} = False$ **then**
18:      $\hat{d} \leftarrow \hat{d} + 1$
19:    **end if**
20: **end while**
21: **Phase 2: Finding the parents, knowing $d$ but not $b_{\min}$.**
22: $\hat{b}_{\min} \leftarrow 1$
23: **for** $i \in T$ **do**
24:    $S \leftarrow$ nodes in $T$ before $i$
25:    **for** $J \subseteq S, |J| = \min(|S|, d)$ **do**
26:      $\text{PASSED} \leftarrow True$
27:      **for** $K \in S\backslash J, |K| \leq \min(|S| - |J|, d)$ **do**
28:        $\beta_{J\cup K} \leftarrow$ Regress $X_i$ on $X_{J\cup K}$
29:        **if** $\exists k \in K : |\hat{\beta}_k| \geq \hat{b}_{\min}$ or $|\hat{\beta}_k| \leq \hat{b}_{\min}/40$ **then**
30:          $\text{PASSED} \leftarrow False$, **break.**
31:        **end if**
32:      **end for**
33:      **if** $\text{PASSED} = True$ **then**
34:        $\hat{\beta} \leftarrow$ regress $X_i$ on $X_J$
35:        $\text{pa}(i) \leftarrow \{j \in J : |\hat{\beta}_j| \geq \hat{b}_{\min}\}$ , **break.**
36:      **end if**
37:    **end for**
38:    **if** $\text{PASSED} = False$ **then**
39:      $\hat{b}_{\min} \leftarrow \hat{b}_{\min}/2$,go to line 23.
40:    **end if**
41: **end for**

---

**For unknown $b_{\min}$ and $d$.** Without knowing $b_{\min}$, we need to consider the following issue: because of the noise, the strength of the edge could be arbitrarily small. Therefore, we could never distinguish the case with faint signal versus no signal. Therefore, when sweeping through the possible values of $b_{\min}$, we may enter a region where all the detected edges have strength more than $b_{\min}$, and the rest have much less. We may not infinitely search for $b_{\min}$. Our solution is to change the algorithm as Algorithm 4.

We first explain what is changed from Algorithm 1. In Phase 1, we add a variable FOUND to determine whether the current estimate $\hat{d}$ is enough to explain the variance of the next node in the ordering. We changed the threshold for MSE in line 11 so that it does not include $d, b_{\min}$. If we are provided sufficient samples, by Proposition A.1, we take the next node with small MSE. If no node has small enough MSE, then FOUND is False, and there is no next node. We can conclude that the number of neighbors $\hat{d}$ is not sufficient to make MSE small enough (there are missing parents), so we need to increment $\hat{d}$ to $\hat{d} + 1$ to fit.

In Phase 2, we changed the threshold in line 29. In line 29, we need to find all the edges to have strength either $\hat{b}_{\min}$ or $\leq \hat{b}_{\min}/40$ for some $K$. If we are given a sufficient number of samples, then with high probability, the ground truth for all those weights is either $\hat{b}_{\min}/2$ or $\leq \hat{b}_{\min}/20$. If for some $\hat{b}_{\min}$ we cannot find such a $K$ (so PASSED is False), it means that even if $J \cup K$ covers all the parents, the coefficients are not within that range. Therefore, we need to make $\hat{b}_{\min}$ smaller and try the algorithm again.

We can perform Phase 1 of Algorithm 4 without knowing $d$ and $b_{\min}$. However, we need $d$ to perform Phase 2 of Algorithm 4, but not $b_{\min}$. And we may change our guarantee to the following, which is comparable to Theorem 2.1. Notice that the number of samples doesn't change in Informal Theorem E.2 and only slightly increases in Informal Theorem E.3.

**Informal Theorem E.2.** Suppose we run Algorithm 4 using $m$ independent samples generated from a DAG $G$ according to (17). If Assumption 1.1 holds, then there exists an absolute constant $C > 0$, such that the following guarantees hold. Phase 1 of Algorithm 1 succeeds in finding a correct topological ordering of the nodes with probability at least $1 - \delta$, provided that

$$m \geq C \frac{1}{b_{\min}^4} \left( d \log(n/d) + \log(1/\delta) \right)$$

**Informal Theorem E.3.** Suppose we run Algorithm 4 using $m$ independent samples generated from a DAG $G$ according to (17). For a given node, we define a **good** set of parents as follows: there is a $b > 0$, such that for all reported $i \to j$, $|b_{ij}| \geq b$ and for other not reported $b_{ij}$, $|b_{ij}| \leq b/10$. If Assumption 1.1 holds, then there exists an absolute constant $C > 0$, such that the following guarantees hold. Provided that Phase 1 of Algorithm 1 succeeds in finding a correct ordering, Phase 2 finds the good set of parents for each node with probability at least $1 - \delta$, provided that

$$m \geq C \frac{\tau(G)}{b_{\min}^2} \left( d \log(n/d) + \log(\frac{1}{\delta \log(1/b_{\min})}) \right)$$

We give a sketch for proving the algorithm's correctness.

1. **Sketch proof of Theorem E.2.** From Lemma A.3, we know that the ratio between MSE and the variance is $\frac{1}{m} \chi_m^2$. Therefore, the largest deviation should be $O(\sqrt{d \log n/m})$, which can be a rubric for the conditional variance gap (i.e., if it is not the next node, then the conditional variance is $O(b_{\min}^2)$, which is $O(\sqrt{d \log n/m})$, by Proposition A.1). Therefore, with high probability, this algorithm will terminate when $\hat{d} = d$ and finally find the true parents. The running time is at most $d$ times the Phase 1 of the original Algorithm 1.

2. **Sketch proof of Theorem E.3.** Since we are given a sufficient number of samples, by the proof of Lemma A.5 and Corollary A.6, we can derive that if $J \cup K$ contains all the parents, then with high probability, the error of $\beta_j - \hat{\beta}_j$ is always $\leq O(b_{\min})$. Then, if we have all weights either $\hat{b}_{ij} \geq b$ or $\hat{b}_{ij} \leq b/40$, then with high probability $|b_{ij}| \geq b - b_{\min}/80$ or $|b_{ij}| \geq b/40 + b_{\min}/80$, which is $|b_{ij}| \geq b/2$ or $|b_{ij}| \leq b/20$, so this achieves our guarantee. Notice that this algorithm terminates when $\hat{b}_{\min} < b_{\min}/2$, so the number of trials of $b_{\min}$ will be $O(\log(1/b_{\min}))$, and by union bound, we can have the algorithm succeed.

**E.4. Extension to Sub-Gaussian Noise.**

One further extension of the model is that we relax the structure of the noise $\varepsilon_i$: the noise is sub-Gaussian. We assume that for all the samples $(X_1^{(k)}, X_2^{(k)}, \ldots, X_n^{(k)})$, the model satisfies:

$$X_i^{(k)} = \sum_{j \in \pi(i)} b_{ji} X_j^{(k)} + \varepsilon_i^{(k)}, \tag{17}$$

Where all the noise $\varepsilon_i^{(k)}$ are independent. Furthermore, for all $i$, the noise $\varepsilon_i^{(1)}, \varepsilon_i^{(m)}$ are independent sub-Gaussian matrices, where $\mathrm{Var}[\varepsilon_i^{(k)}] = 1$ and the sub-Gaussian norm is no more than $\psi_i$. Now, we show that we can perform Phase 1 in 1.

We start with the proof of Theorem 2.1 in Section A. To finish the proof of finding the ordering, we take the same proof by measuring $VE$ and $BE$. It boils down to proving that with this number of samples, $VE$ and $BE$ are upper bounded by $VE \le (1 + O(b_{\min}^2))\mathrm{Var}[X_i|x_J]$ and also $BE \le O(b_{\min}^4)\mathrm{Var}[X_i|x_J]$. The correctness can be guaranteed by Lemmas A.2 and A.3, so it just suffices to prove these two lemmas when Gaussian noise is substituted by sub-Gaussian noise, and the inequality holds for some different $C_V$ and $C_B$.

Before that, we use the following Hanson-Wright Inequality (Rudelson & Vershynin, 2013).

**Lemma E.4** (Hanson-Wright). *Let $X = (X_1, \ldots, X_n) \in \mathbb{R}^n$ be a random vector with independent components $X_i$ which satisfy $\mathbb{E}[X_i] = 0$ and $\|X_i\|_{\psi_2} \le K$. Let $A$ be an $n \times n$ matrix. Then, there is a universal constant $c$ such that for every $t \ge 0$,*

$$\mathbb{P}[|X^\top A X - \mathbb{E}[X^\top A X]| > t] \le 2\exp\left[-c\min\left(\frac{t^2}{K^4\|A\|_F^2}, \frac{t}{K^2\|A\|_{op}}\right)\right].$$

**Proof of Lemma A.2.** We know that $\hat{\beta} - \beta^*$ is a multivariate zero-mean Gaussian with covariance matrix $(\sum_{j=1}^m X_J^{(j)} X_J^{(j)\top})^{-1}$. This follows from the closed-form solution of OLS. So we know that

$$\sum_{j=1}^m \left((X_J^{(j)})^\top(\beta_J^* - \hat{\beta}_J)\right)^2 = (\beta_J^* - \hat{\beta}_J)^\top \left(\sum_{j=1}^m X_J^{(j)} X_J^{(j)\top}\right)(\beta_J^* - \hat{\beta}_J).$$

Let $X$ be the matrix obtained by vertically stacking $X_J^{(1)}, X_J^{(2)}, \ldots, X_J^{(m)}$, and let $Y$ be the column vector $Y = (X_i^{(1)}, X_i^{(2)}, \ldots, X_i^{(m)})$. The error vector $\varepsilon$ is also a column vector, $\varepsilon = (\varepsilon_i^{(1)}, \ldots, \varepsilon_i^{(m)})$. By OLS, we have

$$\hat{\beta}_J = (X^\top X)^{-1}(X^\top Y) = (X^\top X)^{-1} X^\top(X\beta^* + \varepsilon),$$

which implies

$$\sum_{j=1}^m \left((X_J^{(j)})^\top(\beta_J^* - \hat{\beta}_J)\right)^2 = \varepsilon^\top\left(X(X^\top X)^{-1}X^\top\right)\varepsilon.$$

Therefore, let $A = X(X^\top X)^{-1}X^\top$, which is a projection matrix. We know that $\|A\|_F^2 = k$, $\|A\|_{op} = 1$, and $K = \psi_i$. Also, since $\varepsilon_i^{(k)}$ are independent noise with variance 1, we have $\mathbb{E}[\varepsilon^\top A\varepsilon] = k$. Accordingly, we can set

$$m = O\left((\log(1/\delta) + k)\,\psi_i^2/\varepsilon\right),$$

and that suffices.

**Proof of Lemma A.3.** The sum of squares of all random variables is a special case of the Hanson-Wright Inequality (Rudelson & Vershynin, 2013) with $A = I_m$. Therefore, for the expected sum of squares of all $Y_i^{(k)}$'s, we have

$$\mathbb{P}\left[\left|\sum_{i=1}^m Y_i^{(k)2} - mV^2\right| > tV^2\right] \le 2\exp\left[-c\min\left(\frac{t^2}{\psi_i^4 m}, \frac{t}{\psi_i^2}\right)\right].$$

Hence, we know that

$$m = O\left(\log\left(\tfrac{1}{\delta}\right)\frac{\psi_i^2}{\varepsilon^2}\right)$$

suffices.

# F. Further Simulation Results

The data setting is the one described in Section 4. We provide here some more simulation results. We have two configurations of data. We vary different $d, n, b_{\max}, N$ parameters and test the results.

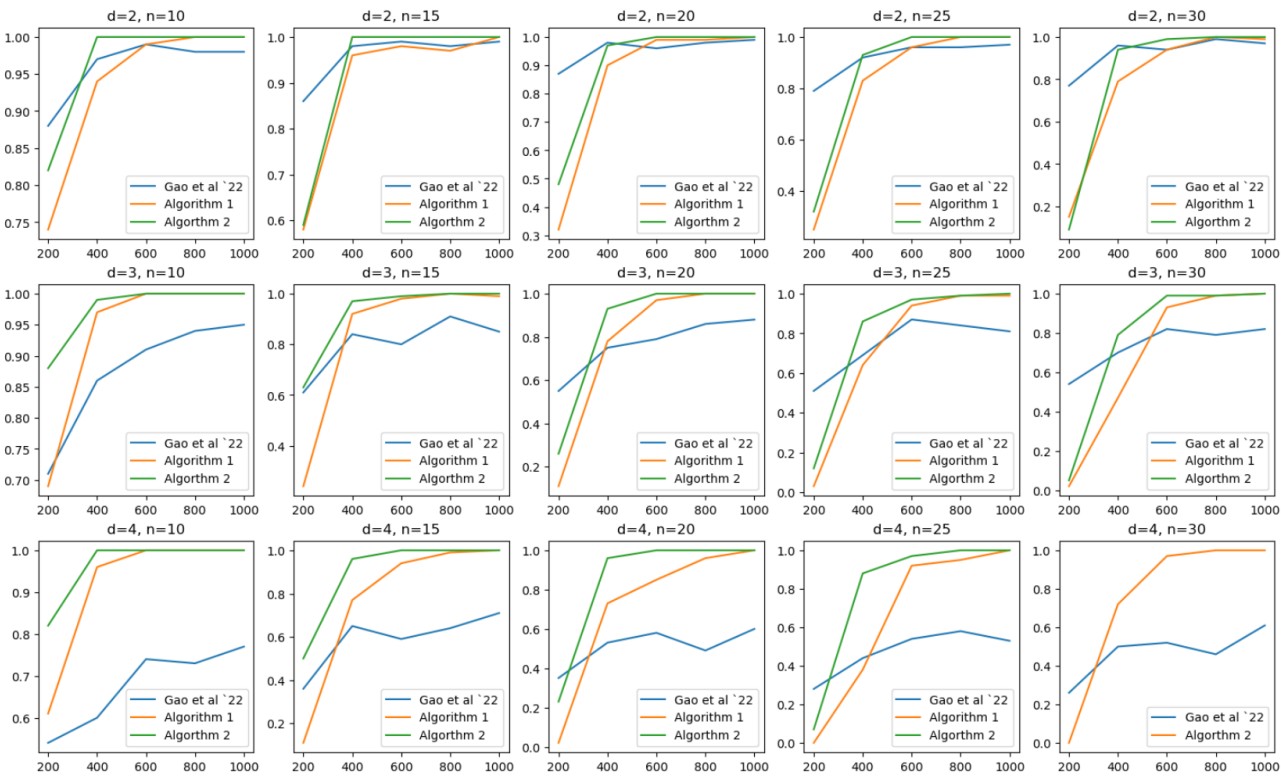

*Figure 7.* Figure for $b_{\max} = 1$

Figure 7 shows a set of simulation results with $b_{\max} = 1$. The test metric is to test 100 different random graphs, and for each graph test for those graphs whether (Gao et al., 2022), Algorithm 1 and Algorithm 2 discover the same topology.

We also performed another set of experiments for $b_{\max} = 2$ and the results are shown in Figure 8.

The test metric is to test 100 different random graphs, and for each graph test for those graphs whether (Gao et al., 2022), Algorithm 1 and Algorithm 2 discover the same topology. Also, we add the number of false positive edges with 95% confidence interval (centered with mean and $\pm 2\sigma$ error region) as in Figure 9. Because of the long runtime, for the case when $d = 4$ and $n = 30$, the experiment of Algorithm 1 is not performed.

For all the cases we can find out that eventually, when the number of samples $m$ grows, Algorithm 1 and Algorithm 2 converge at a faster rate than (Gao et al., 2022). We can see that for $b_{\max} = 1$ case all the algorithms perform better then for $b_{\max} = 2$, which is expected from the theoretical results, since $\tau(G)$ and $R$ grows much faster (as an exponential function) when $b_{\max}$ is larger than 1. For both (Gao et al., 2022) and Algorithm 2, we see a degradation in performance as $n$ increases, while comparatively, Algorithm 1 is not affected as much. This can be explained by the growth of $\tau$, which seems to be sublinear in $n$, in contrast with $R$ which grows linearly and the condition number which grows superlinearly (see also Figure 1).

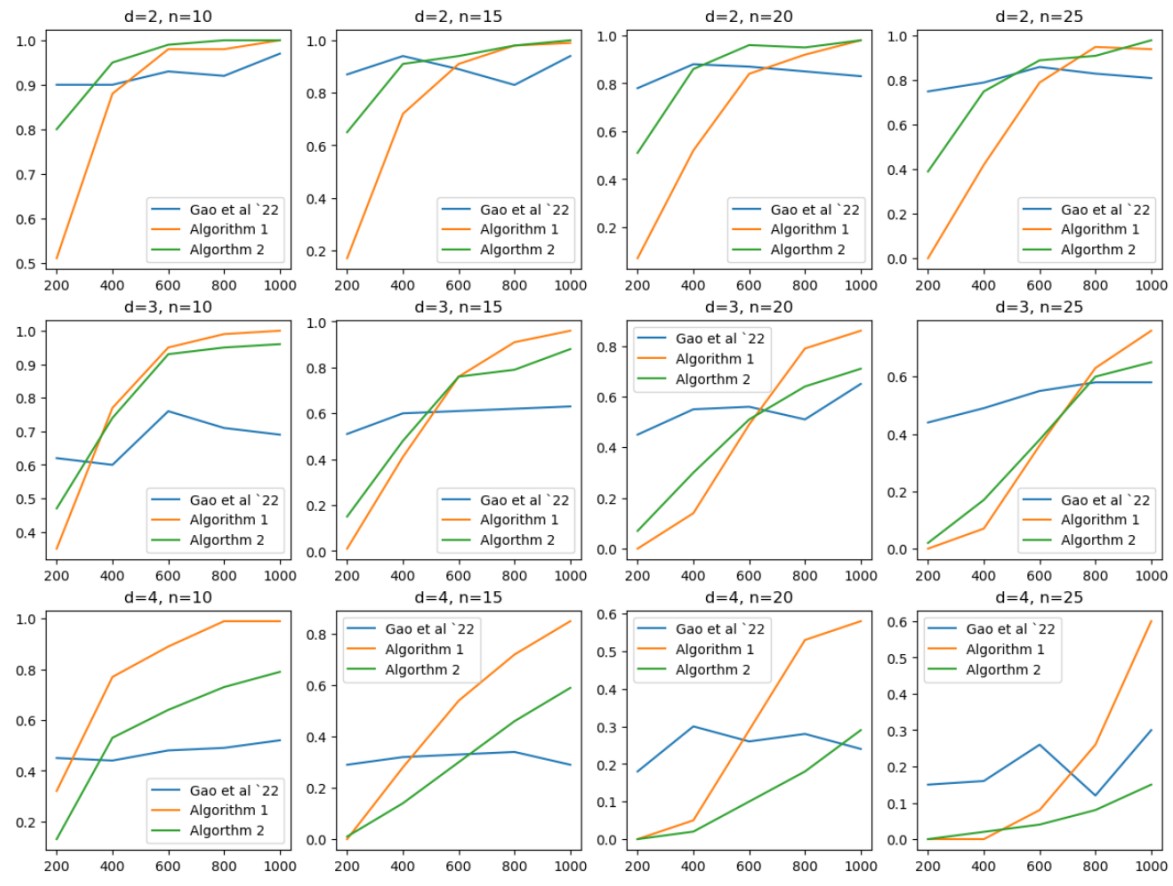

*Figure 8.* Figure for $b_{\max} = 2$

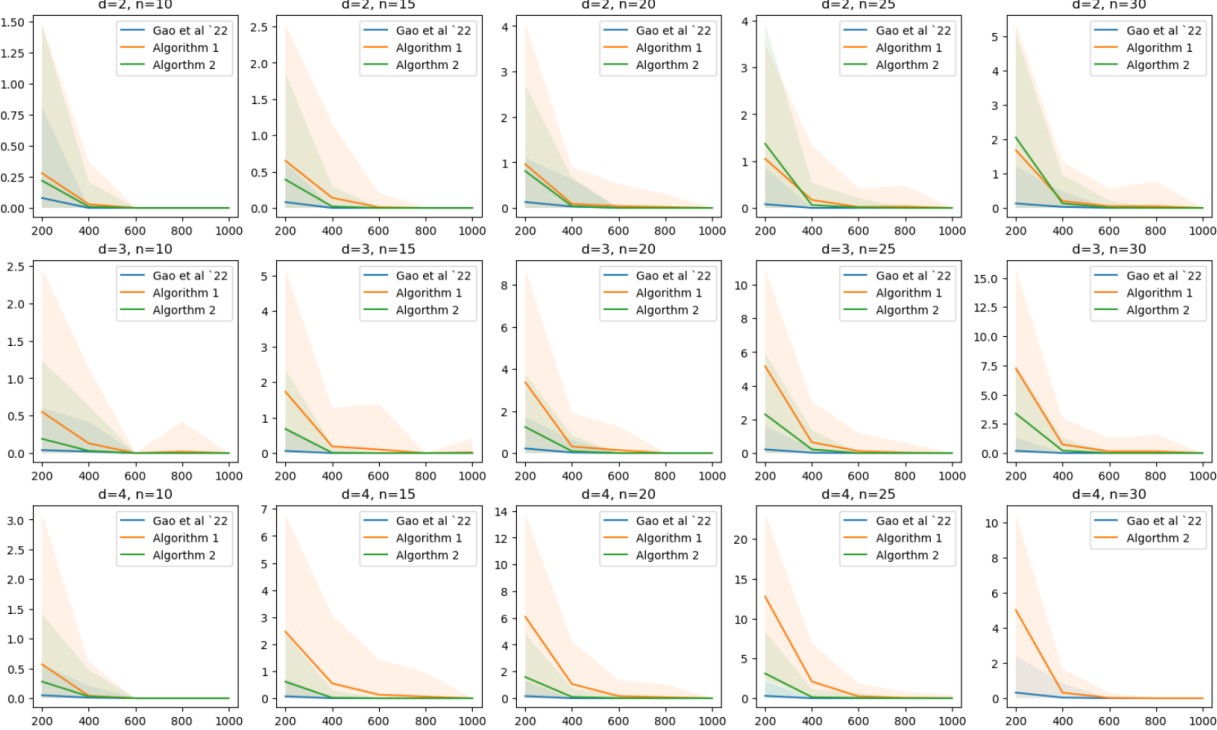

*Figure 9.* Figure for $b_{\max} = 1$

## G. Code

```python
1  import numpy as np
2  from sklearn.linear_model import Lasso
3  from sklearn.linear_model import LinearRegression
4  import random
5
6  def sample_d(lst, d):
7      # Sample <=d things in the list
8      n = len(lst)
9      if n==0: return []
10     parents = []
11     probability = min(d/(n+2),1/2)
12     for node in lst:
13         if random.random() < probability:
14             parents.append(node)
15     return parents[:d]
16
17 def generate_config(n, d, b_min, b_max):
18     # Sample a random Bayesnet condiguration of n vertices with <=d indegree
19     order = list(range(n))
20     random.shuffle(order)
21     coefficient = {i:{} for i in range(n)}
22     parents = {i:set() for i in range(n)}
23     B = [[0]*n for _ in range(n)]
24     for i in range(n):
25         node = order[i]
26         prt = sample_d(order[:i],d)
27         parents[node] = set(prt.copy())
28         for p in prt:
29             abs_value = (b_min+(b_max-b_min)*random.random())
30             sign = (2*random.randint(0,1)-1)
31             bij = abs_value * sign
32             B[p][node] = coefficient[node][p] = bij
33     return order, coefficient, parents, np.array(B)
34
35 def generate_sample(B, N):
36     '''
37     From the sample configuration sample a set of values. N is the number of
38     samples, B is the coefficient matrix.
39     '''
40     n = B.shape[0]
41     raw = np.random.normal(0,1,[N,n])
42     return raw@np.linalg.inv(np.eye(n)-B)
43
44 def combination(lst, d):
45     # List all subsets (as list) with <=d elements in lst
46     if d == 0 or len(lst) == 0:
47         return [[]]
48     else:
49         return combination(lst[1:], d) + [[lst[0]]+j for j in combination(lst[1:], d-1)]
50
51 def choose(lst, d):
52     # List all subsets (as list) with =d elements in lst
53     if d == 0:
54         return [[]]
55     elif d > len(lst):
56         return []
57     else:
58         return choose(lst[1:], d) + [[lst[0]]+j for j in choose(lst[1:], d-1)]
59
60 def lr(data, cand, index):
61     '''
62     Do OLS linear regression, given the data, candidate of nodes (cand) to
63     regress on and the index of the node whom to regress on. Return the
```

```
64      coefficient (if cand is nonempty) and the MSE
65      '''
66      assert index not in cand
67      N = data.shape[0]
68      Y = data[:,[index]]
69      X = data[:,cand]
70      coef = np.linalg.inv(X.T@X)@(X.T@Y)
71      mse = ((X@coef-Y)**2).sum()/N
72      return coef.T, mse
```

*Listing 1.* Data-sampling algorithm

```
1  def topo(data, d):
2      '''
3      Find the topological order given the data and d. Return the topological
4      order as a list of nodes.
5      '''
6      N, n = data.shape
7      rem_nodes = list(range(n))
8      top = [np.argmin(sum(data**2))]
9      rem_nodes.remove(top[0])
10     for _ in range(n-2):
11         best_mse = 1_000_000
12         next_node = -1
13         for r in rem_nodes:
14             mse = 1_000_000
15             for cand in combination(top, d):
16                 coef, mse1 = lr(data, cand, r)
17                 mse = min(mse,mse1)
18             if mse < best_mse:
19                 best_mse = mse
20                 next_node = r
21         rem_nodes.remove(next_node)
22         top.append(next_node)
23     top.append(rem_nodes[0])
24     return top
```

*Listing 2.* Phase 1 of Algorithm 1: topological order

(Gao et al., 2022)

```
1  def learning_parents_g(data, order, d):
2      '''
3      Using Gao'22's algorithm to find the patents given the topological order.
4      Return the partents for each node as a dictionary.
5      '''
6      n = data.shape[1]
7      parents = {i:set() for i in range(n)}
8      for i in range(n):
9          if i==0:
10             continue
11         prev = order[:i]
12         node = order[i]
13         mse = 1_000_000
14         parent_superset = None
15         # Find C_j to be argmin(v_jC)
16         for cand in combination(prev, d):
17             coef, mse1 = lr(data, cand, node)
18             if mse1 < mse:
19                 mse = mse1
20                 parent_superset = cand
21         parent = set()
22         for pa in parent_superset:
23             parentm1 = parent_superset.copy()
24             parentm1.remove(pa)
```

```
25            coef, mse1 = lr(data, parentm1, node)
26            if abs(mse-mse1) >= gamma:
27                parent.add(pa)
28        parents[node] = parent
29    return parents
```

*Listing 3.* Finding the parents using (Gao et al., 2022) algorithm

```
 1 def learning_parents_v(data, order, d):
 2     '''
 3     Using the inefficient algorithm to find the patents given the topological
 4     order. Return the partents for each node as a dictionary.
 5     '''
 6     n = data.shape[1]
 7     parents = {i:set() for i in range(n)}
 8     for i in range(n):
 9         cache = {}
10         if i==0:
11             continue
12         prev = order[:i]
13         node = order[i]
14         parent_superset = None
15         if i <= d:
16             parent_superset = prev
17         else:
18             for J in choose(prev, d):
19                 passed = True
20                 others = list(set(prev)-set(J))
21                 for K in choose(others, min(d,i-d)):
22                     JKsorted = tuple(sorted(J+K))
23                     if JKsorted not in cache.keys():
24                         # print("new", JKsorted)
25                         coefJK, mse1 = lr(data, sorted(J+K), node)
26                         cache[JKsorted] = coefJK
27                         # print(coef)
28                     else:
29                         # print("cache")
30                         coefJK = cache[JKsorted]
31                     cK=[abs(coefJK[0][m]) for m in range(len(J+K)) if JKsorted[m] in K]
32                     if max(cK) >= b_min/2:
33                         passed = False
34                         break
35                 if passed:
36                     break
37             parent_superset = J
38         coef, mse1 = lr(data, parent_superset, node)
39         parent = set(n for (c,n) in zip(coef[0], parent_superset) if abs(c) >= b_min/2)
40         parents[node] = parent
41     return parents
```

*Listing 4.* Finding the parents using the Phase 2 of Algorithm 1

```
 1 def lasso(data, cand, index):
 2     N = data.shape[0]
 3     Y = data[:,[index]]
 4     X = data[:,cand]
 5     Y1 = data[:,[index]]
 6     X1 = data[:,cand]
 7     lambda_n = 0.01
 8     if cand == []:
 9         return [], (Y**2).sum()/Y.size
10     reg = Lasso(fit_intercept = False, alpha = lambda_n, copy_X = False).fit(X,Y)
11     coef = reg.coef_
12     return coef, ((X@np.array([coef]).T-Y)**2).sum()/N
```

```
13
14  def learning_parents_e(data, d):
15      # Learn the parents by LASSO, the efficient algorithm
16      N, n = data.shape
17      rem_nodes = list(range(n))
18      top = [np.argmin(sum(data**2))]
19      rem_nodes.remove(top[0])
20      parents = {i:set() for i in range(n)}
21      for _ in range(n-1):
22          best_node = -1
23          parent = set()
24          best_mse = 1_000_000
25          for node in rem_nodes:
26              coef, mse = lasso(data, top, node)
27              if mse <= best_mse:
28                  best_mse = mse
29                  best_node = node
30                  parent = set(n for c, n in zip(coef,top) if abs(c) > b_min/2)
31                  parent = set(list(parent)[:d])
32          parents[best_node] = parent
33          top.append(best_node)
34          rem_nodes.remove(best_node)
35      return top, parents
```

*Listing 5.* Algorithm 2: the topological order and parent

```
1  n = 5 # Choose any n you want
2  d = 2 # Choose any d you want
3  b_min = 0.5
4  b_max = 1 # Choose any b_max you want
5  sigma = 1
6  gamma = b_min**2/2 # As Gao'et al 2022
7  repeat = 45 # Choose the number of trials you want
8
9  def doing_regular(tup):
10     '''
11     Genetare a process to perform all three of those algorithm
12     tup is of 2 integers (j, N) where j is a dummy parameter and N is the
13     number of the samples.
14     Return the result of whether they have find the correvt parents for three
15     algorithm.
16     '''
17     j, N = tup
18     if j%20 == 0: print(j)
19     order, coefficient, parents, B = generate_config(n, d, b_min, b_max)
20     data = np.array(generate_sample(B, N))
21     order = topo(data,d)
22     g = learning_parents_g(data, order, d) == parents
23     v = learning_parents_v(data, order, d) == parents
24     e = learning_parents_e(data, d)[1] == parents
25     return g, v, e # Gao et al's, inefficient, efficient
```

*Listing 6.* Paremeter settings and function to do the experiments

