# OpenReview forum: "Learning Gaussian  DAG Models without Condition Number Bounds"
_ICML.cc/2025/Conference — ICML 2025 poster_

### Official Review · Reviewer_TgLM · 2025-03-08

**Overall Recommendation:** 4

**Summary:**

In this paper, the authors revisit the problem of learning an $n$-variate Gaussian graphical model from samples. This problem is known to be solvable in $O(d\log n)$ samples where $d$ is the degree. However, there is a hidden polynomial dependence on the condition number of the covariance matrix of observations which can be polynomial in $n$ in the worst case.

Instead, the authors make an assumption that the sum of squares of the linear SEM coefficients ($\tau$ parameter) of a node is bounded. Together with this, they also have the usual assumptions such as each noise is standard Gaussian and the coefficients are not too close to 0 ($b_{min}$ parameter). Under this assumption, they manage to give an algorithm that runs in $O(n^{2d+2})$ time and $O(d\log n)$ samples. They also show using Fano's method that their dependence on the previous two parameters is also optimal. Although, there is a gap of $d$ between the upper and lower bounds.

Subsequently, they give an alternate algorithm that runs in poly(n,d) time, but its sample complexity is worse than that of the first algorithm. They further show that their first assumption is weaker that the condition number assumption. Technically, their algorithm is split is two phases: 1) topological order recovery using a greedy method that picks the lowest variance first  2) parent set recovery using regression using an existing result for undirected graphs.

The authors perform experiments with synthetic data to confirm their findings. The results show that their accuracy for graph recovery is better than the existing methods once the sample size becomes large.

**Claims And Evidence:**

Yes.

**Essential References Not Discussed:**

NA

**Experimental Designs Or Analyses:**

It would have been more convincing if they also had experimental results for benchmark datasets along with synthetic datasets.

**Methods And Evaluation Criteria:**

Yes.

Although, it looks to me too few (only one) prior work has been compared against. Section E in the appendix had some more such comparisons. Maybe some of it could be brought in the main body.

**Other Comments Or Suggestions:**

NA

**Other Strengths And Weaknesses:**

In section 2.4 the authors interpret their theoretical results in comparison with the existing literature. It would have been also nice if they can point out their main technical novelty that gave them improved experimental result. What is the new thing they did algorithmically that gave the improvement? If they discuss this in more detail, it will make the paper more understandable.

**Questions For Authors:**

see above

**Relation To Broader Scientific Literature:**

I think the claim that $\tau$ is a better assumption has been theoretically justified. The proposed algorithms is also performing better empirically.

**Theoretical Claims:**

I briefly looked at section A and B. It looks okay at a quick glance. I didn't go through the proofs in detail.

---

> ### Author Rebuttal · Authors · 2025-04-01
>
> We thank the reviewer for their detailed response and thorough examination of our work. We answer the main points that were raised below.
>
> >Although, it looks to me too few (only one) prior work has been compared against. Section E in the appendix had some more such comparisons. Maybe some of it could be brought in the main body.
>
> The reviewer is absolutely correct, the manuscript would benefit from some further discussion of prior work in the main body. We will make sure to bring some of this discussion earlier instead of only appearing in the appendix.
>
> >It would have been more convincing if they also had experimental results for benchmark datasets along with synthetic datasets.
>
> Thank you for raising this point. We agree that including benchmark datasets in the simulations would complement the theoretical picture even more. Our focus in this work was to theoretically establish the optimal sample complexity for this problem and to demonstrate how that translates to improvements over previous algorithms. Since the algorithms we were comparing against were tested on synthetic datasets, we made the same choice.
>
> >In section 2.4 the authors interpret their theoretical results in comparison with the existing literature. It would have been also nice if they can point out their main technical novelty that gave them improved experimental result. What is the new thing they did algorithmically that gave the improvement? If they discuss this in more detail, it will make the paper more understandable.
>
> Thank you for raising this important point. We will make sure to clarify the experimental improvement in the final version.  Our algorithm for finding the parents of a given node $X_i$ is based on a connection that we observed between our problem and that of learning undirected gaussian graphical models. Thus, it proceeds by choosing a candidate neighborhood $S$ of size $d$ and then regressing $X_i$ with $X_{S\cup T}$, where $T$ ranges over all other possible neighborhoods of size $d$. It only accepts a candidate neighborhood $S$ as the true one if for all these regressions we never find a node outside $S$ with large coefficient. In contrast, the previously known algorithm chooses the subset $S$ with the smallest Mean Squared Error (MSE) in the regression of $X_i$. This is a superset of the true neighborhood, so they find all nodes in $S$ such that removing them from $S$ increases the MSE by less than $\gamma$ and delete them. Unfortunately, their choice of $\gamma = \Theta(b_{\min}^2)$ is too big, resulting in many true parents being deleted. In our analysis, we quantify the correct threshold for deletion and show that it also depends on $\tau$. This difference has a significant impact on performance.

---

> > ### Comment · Reviewer_TgLM · 2025-04-07
> >
> > Based on the authors' response I am more convinced about the strength of the paper. So, I am updating my score accordingly.

---

### Official Review · Reviewer_vPQm · 2025-03-13

**Overall Recommendation:** 4

**Summary:**

This submission investigates the estimation of Gaussian DAG structure from i.i.d. samples and under equal-variance assumption. The main findings are two-fold: the authors give a polynomial time algorithm to recover the structure based on ideas of sparse regression; the authors shows that the sample complexity can be expressed in terms of the maximum sump of squares of the coefficients for the out-edges. Furthermore, they show that this complexity parameter improve upon standard analyses based on conditional number, as this latter may grow polynomially with the dimension. Hence, their method is particularly interesting in high-dimensional settings where the number of nodes of the graph is large.

**Claims And Evidence:**

The submission developed the theoretical aspects under reasonable assumptions. The authors proved upper and lower bounds, and sample complexity to support their analysis. Numerical experiments on synthetic data confirms the benefits of their method in high-dimensional settings.

**Essential References Not Discussed:**

Not to my point of view.

**Experimental Designs Or Analyses:**

I did not run their code but I believe that their reported experiments are sounds.

**Methods And Evaluation Criteria:**

Evaluated on synthetic data

**Other Comments Or Suggestions:**

Typos:
024 space between : PC-algorithm(Spirtes et al., 2001)
070 space between : model selection(Spiegelhalter
other spaces between word+ref, please check.
149 variance should be one: \sigma^2=1, wlog
368 \in missing : jpa(i)

**Other Strengths And Weaknesses:**

The submission is well written and delivers a quite exhaustive analysis going from information theoretic result to polynomial time algorithm with guarantees.

**Questions For Authors:**

.

**Relation To Broader Scientific Literature:**

This paper investigates DAG models, applications of DAGs can be interested in these results and algorithms.

**Theoretical Claims:**

I have checked the sample complexity of Algorithm 2 (Theorem 2.6), various lemmas in the main document.

---

> ### Author Rebuttal · Authors · 2025-04-01
>
> We thank the reviewer for their encouraging feedback and for pointing out these omissions, we will make sure to correct them in the final version.

---

### Official Review · Reviewer_QEWp · 2025-03-14

**Overall Recommendation:** 4

**Summary:**

This paper studies the sample complexity of learning linear Gaussian DAGs under equal variance assumption. It proposes a new algorithm and provides the graph recovery guarantee independent of the condition number. Both the upper and lower bounds of sample complexity are proved. The authors also provide simulation to verify the proposed method and demonstrate improvement compared to prior results.

**Claims And Evidence:**

Yes

**Essential References Not Discussed:**

Not that I am aware of.

**Experimental Designs Or Analyses:**

Yes, I checked the simulation setting. The setup seems reasonable.

**Methods And Evaluation Criteria:**

Yes, it is evaluated on simulated dataset. The data generation process of the simulated data and the evaluation criteria makes sense to me.

**Other Comments Or Suggestions:**

1. For figure 2, it would also be good to include the error bar.
2. In Algorithm 1, it would be clearer to state how T is sorted and how to handle the topological order when two nodes have same MSE.
3. For the experiments, it would be useful to report the TPR and FPR of the edge discovery.

**Other Strengths And Weaknesses:**

1. This paper is clearly written and the problem it studies is interesting and contributes to the theoretical aspect of graphical models.
2. Algorithm 1 suffers from computational complexity. Algorithm 2, while more efficient, has some loss in sample complexity.

**Questions For Authors:**

1. To run Algorithm 1,2, knowing $b_min$ is required. How is this value estimated in practice?
2. In Lemma 2.2, the authors showed that $\tau(G)\leq \kappa$. Is there an example that $\tau(G)$ has an order of magnitude smaller than $\kappa$? Since Theorem 2.1 depends on $\tau(G)$. The theorem would be more convincing if there is an example demonstrating that $\tau(G)$ is of magnitude smaller than $\kappa$.
3. The simulation results are in the low-dimensional regime. It would be nice to see the result in higher dimensional setting (when the ratio of n/m is smaller, and n is larger).

**Relation To Broader Scientific Literature:**

Learning graphical models can help drive deeper understandings for the physical process of biology, neuroscience and so on. Applications in this domain often have limited sample size. Hence, understanding the sample complexity of the algorithm helps better assess the quality of the estimation and understand its limitation.  Furthermore, this paper proposes a new algorithm that is shown to recover the graph without dependence in condition number, which often grows with respect to the number of parameters. This makes estimation in high-dimensional setting with small sample size possible.

**Theoretical Claims:**

I checked the proof of Lemma 2.2. It seems reasonable.

---

> ### Author Rebuttal · Authors · 2025-04-01
>
> We thank the reviewer for their encouraging words and detailed feedback. Below we answer the main points that were raised.
>
> # For comments:
>
> >For figure 2, it would also be good to include the error bar.
>
> Thank you very much for this suggestion. We will make sure to include the error bars in the final version.
>
> >In Algorithm 1, it would be clearer to state how T is sorted and how to handle the topological order when two nodes have the same MSE.
>
> Thank you very much for this question, it is very important to clarify how $T$ is updated since that might not be clear from the pseudocode. Essentially, $T$ is a list of nodes that is initialized as empty and every time we find the node with the smallest MSE we append it at the end of the list. If two nodes happen to have the same (smallest) MSE, then we choose either of them to append to $T$. This is because having the same MSE indicates they are both valid to be the next node in the topological ordering since they have the same conditional variance given the previous nodes, so we can pick any of them as the next node.
>
> >For the experiments, it would be useful to report the TPR and FPR of the edge discovery.
>
> Answer: We agree this would be an excellent benchmark for performance, we will make sure to report these in the final version of the manuscript.
>
> # For questions:
>
> >To run Algorithm 1,2, knowing $b_{\min}$ is required. How is this value estimated in practice?
>
> We thank the reviewer for this insightful question. If we interpret the question correctly, this question is about how the algorithm would operate without knowing $b_{\min}$ beforehand. We have included a detailed discussion in appendix E.3 about adaptivity when various parameters (including $b_{\min}$) are unknown. We will make sure to make this discussion more visible in the main text.
>
> Essentially, without knowing $b_{\min}$, the algorithm would proceed with an initial estimate $b$ of $b_{\min}$. Then, we run the Algorithm assuming $b_{\min} = b$ and in the end, we check whether the estimated strengths of all the edges are either less than $b/10$ or larger than $b$. If we detect some edge that has strength between $b/10$ and $b$, we half $b$ and run the process again. We can show that if our number of samples is the one specified by our main Theorems for Algorithms 1 ,2, then such a process will eventually terminate and return a $b$ such that all edges are either smaller than $b/10$ or larger than $b$.
>
> >In Lemma 2.2, the authors showed that $\tau(G)\le\kappa$. Is there an example that $\tau (G)$ has an order of magnitude smaller than $\kappa$? Since Theorem 2.1 depends on  $\tau (G)$. The theorem would be more convincing if there is an example demonstrating that  $\tau (G)$ is of magnitude smaller than $\kappa$.
>
> Thank you very much for raising this point, we absolutely agree that such an example would help demonstrate the improvement over previous bounds. Such an example is essentially provided in Lemma 2.5 (ii), where the topology is a binary tree, where each edge has $2^{-1/4}$ weight (which makes $\tau$ a constant), but the condition number grows as $\kappa = \Omega(\sqrt{n})$. We will make sure to highlight this example immediately after presenting Theorem 2.1, so that it is clear to the reader why the dependence on $\tau$ is preferred over $\kappa$.
>
> >The simulation results are in the low-dimensional regime. It would be nice to see the result in higher dimensional setting (when the ratio of n/m is smaller, and n is larger).
>
> Thank you for pointing out. We will make sure to include simulations with higher $n$ that better reflect the high dimensional nature of the problem.

---

### Decision · Program_Chairs · 2025-05-01

**Decision:**

Accept (poster)

**Comment:**

The paper studies learning of Gaussian linear structural equation models under equal variance. Algorithms are proposed. For n nodes and degree d, theoretical analysis shows that the sample complexity (lower and upper bound) is O(d log n) which has been observed before in Gao'22. The truly novel result is that the bound is now independent of the condition number.

The reviewers agree to accept this paper, although the time complexity for the proposed algorithm is O(n^(2d+2)). A more efficient LASSO estimator is provided, but for the new algorithm the sample complexity is O(d^4 log n).

Please take into the account the comments from all reviewers regarding for instance experiments with larger number of nodes n, or point out the main technical novelty that allowed for improved experimental results.